# Performance Analysis of Ground Target Detection Utilizing Beidou Satellite Reflected Signals

**DOI:** 10.3390/s19092163

**Published:** 2019-05-09

**Authors:** Chaoqun Gao, Dongkai Yang, Xuebao Hong, Bo Wang, Bo Zhang

**Affiliations:** School of Electronic and Information Engineering, Beihang University, Beijing 100191, China; 13370149796@163.com (C.G.); yangdongkai@sina.com (D.Y.); Joyce_hong2008@yeah.net (X.H.); buaamrwang@163.com (B.W.)

**Keywords:** BeiDou, reflected signals, target detection, phase difference information, I branch component, CNR

## Abstract

This paper presents a method of ground target detection using reflected signals of BeiDou satellites. The phase difference information, which is the output of the phase-lock loop (PLL) in the tracking process, is an important observation in this technique. The geometric relationships between the specular point of different BeiDou satellites and the target are established. In addition, the detection and false alarm probability are also analyzed. In order to verify the reliability of the method, an experiment in the suburb area of Beijing was completed. The target was placed in the coverage area of the left-handed circular polarization (LHCP) antenna for two time periods (10–20 s and 40–55 s). By observing the phase difference in BeiDou reflected signals in the presence of a target, it was found that the changing trend was in good agreement with the target placement time periods. In the second experiment, the target moved east and west at a speed of 0.5 m/s, and the range of motion was 6 m. During the acquisition of the BeiDou reflection signal, the target passed through the antenna 14 times. The performance of target detection with different parameters was observed by extracting in-phase (I) branch component data, phase difference information, and the carrier-to-noise ratio (CNR) of five BeiDou reflected signals. The experimental results allowed three conclusions to be drawn as follows: (1) The target detection performance of the three parameters has a certain relationship with the altitude angle and the azimuth angle of the satellite; (2) target motion direction information can be reflected in the change of the satellite I branch component data; (3) The CNR information of different satellite reflected signals varies greatly when the target moves, which is quite different from that of the first experimental target when it is stationary. Thus, the feasibility of target detection using BeiDou reflection signal was demonstrated through these two experiments.

## 1. Introduction

Global navigation satellite system (GNSS) reflectometry has been widely used in the remote sensing field because of its low cost, all-weather capability, and global coverage [1,2]. Good results have also been achieved in sea ice detection [3,4], ocean altimetry [5,6], sea wind retrieval [7,8], soil moisture monitoring [9,10], oil slicks [11], and dry snow detection [12].

The P(Y) and C/A ReflectOmeter (PYCARO), which flew on-board a stratospheric balloon, has been exploited to measure the scatterers’ height fluctuations using the peaks of reflected complex waveforms [13]. In addition, the difference in the coherent-to-incoherent scattering ratio over boreal forests and lakes has also been discussed. The signal scattering over ocean surface from GPS satellites have been discussed in [14]. In the case of smooth surface, the signal-to-noise (SNR) ratio and phase information will follow the Fresnel coefficients. In the case of rough surface, the signal scattered similarly at vertical polarization and horizontal polarization. Based on it, the results of polarimetric ratio over the sea ice has good performance. At the same time, the scattering properties over land have also been evaluated using SNR information, the polarimetric ratio, and the widths of the waveforms’ trailing and leading edges [15]. During the balloon experiments for university students (BEXUS), the presence of 17 stratospheric balloons, which are strong coherent components of the forward scattered signal, was proven [16]. Based on the multimodal behavior of received power, the coherent scattering component from a forest was analyzed. The variability of the coherent phase samples and polarimetric measurements was compared with in situ observations to make a realistic rough characterization of the ice cover [17]. The signals reflected from this were used to monitor the complete process of sea ice formation and melting of Greenland during a seven-month period in 2012.

However, most of these studies used GPS and Galileo systems, whose satellites move at the medium Earth orbit (MEO). The BeiDou system can not only provide the signal of the MEO satellite, but also the signals of the inclined geosynchronous orbit (IGSO) and the geostationary Earth orbit (GEO) satellites, which can provide stable geometry and better coverage capability in the mid- and low-latitude regions [18,19]. With the completion of the global network of the BeiDou system in recent years, more and more research has been done in the field of remote sensing using its reflected signals. The BeiDou reflectometry software receiver has been exploited in two coastal experiments on Dishui Lake and Dayangshan Port [20], whose resulting error was 0.11 m. Coastal Typhoon observations were completed by using the ocean reflected signals from BeiDou GEO satellites [21]. The relationship between reflected waveform parameters, such as coherent time, and the ocean wind speed were analyzed in [22]. After comparing the in situ wind measurements collected during two tropical cyclones, it was confirmed that the BeiDou reflected signals can retrieve wind speed effectively. Besides that, the feasibility of coastal ocean phase altimetry and sea state observation in the frequency domain with BeiDou GEO reflected signals was proven in [23]. Because of the stable elevation of the BeiDou GEO satellite, two experiments on coastal sea ice detection were performed in Bohai Bay in China from 30 January to 4 February 2016. The results showed that the concentration of sea ice is correlated with the polarization ratio of BeiDou GEO satellites [24]. The same application was also tested in [25]. The soil moisture retrieval method was proposed based on the reflected signals from BeiDou GEO satellites, using a support vector regression machine (SVRM)-assisted method. It was found that BeiDou satellites are appropriate for use in this field [26]. The literature [27] has also investigated the feasibility of soil moisture estimation based on BeiDou B1 band interference signals. Up until now, sea ice detection, sea breeze monitoring, soil moisture, and other research areas have been based on extracting effective information from reflected signals, such as phase information, interference signals, and the polarization ratio to invert different media on the Earth’s surface. Because of the difference in the reflection medium between the target and the ground, the target on the ground can be effectively detected by using the reflected signal. The principle of using phase difference information in the tracking loop of GPS-reflected signals to detect ground targets was introduced in detail in [28].

The paper further proposes the use of the reflected signal of the BeiDou satellite to detect ground targets, and carries out the relevant experiment. Section 2 briefly describes the principle of ground detection based on the phase difference information in the tracking loop. Additionally, the relationship between the carrier-to-noise ratio (CNR) information and the target presence is presented. In Section 3, two experiments that were conducted in the suburban area of Beijing city are introduced, including the instruments used, the target movement, and the data collection. Section 4 presents the data processing results of the two experiments in detail, and the results are analyzed based on the geometric relationships among the specular points and the reflection mechanism of the signals. Finally, the conclusions are presented.

## 2. Detection Method

### 2.1. Phase Difference Analysis of Reflected Signal

When aiming to extract the navigation message from the GNSS satellite, the carrier of the signal should be removed. To accomplish this goal, the receiver needs to adjust the code phase and the carrier frequency of locally generated replicas to align with the corresponding satellite signal. In the process, the carrier phase can be estimated by the phase locked loop (PLL) in the tracking loop [29]. PLL uses a feedback function to control the behavior of the numerically controlled oscillator (NCO), which can guarantee that the replica signal is accurate. The core role of the feedback loop is to measure the differences in frequency and phase among the received signals and the locally generated replicas [30]. Based on these differences, the controller commands the NCO to increase or decrease its frequency. In the ground target detection application, information about the target was carried in the reflected signal. The phase difference information in the PLL can be used to detect it effectively. In the BeiDou software receiver used in this paper, the second-order Costas loop was used for the tracking operation, which is shown in Figure 1.

In the absence of a ground target, the reflected signal received by the left-handed circularly polarized (LHCP) antenna came from the ground, whose geometric relationship is shown in Figure 2. In this case, the input signal of PLL $u_{i_{1}}$ can be expressed as
(1)$$u_{i_{1}}=u_{ref-road}\left(t\right).$$

After mixing the input signals and the locally generated replica signals, the low pass filter was used to filter out the high frequency components contained in the mixing results. Its in-phase (*I*) branch and quadrature (*Q*) branch are expressed as
(2)$$I_{P}\left(t\right)=aD\left(t\right)\cos\left(w_{e}t+\theta_{e}\right)$$
and
(3)$$Q_{P}\left(t\right)=aD\left(t\right)\sin\left(w_{e}t+\theta_{e}\right),$$
where $w_{e}$ and $\theta_{e}$ are the frequency and initial phase differences between the input signals and the locally generated replicas, respectively. According to the above two formulas, two signals, the *I* branch and the *Q* branch, in the phase discriminator can also be expressed as
(4)$$\gamma_{P}\left(t\right)=I_{P}\left(t\right)+jQ_{P}\left(t\right)=aD\left(t\right)e^{j(w_{e}t+\theta_{e})}.$$

The phase difference, $\Delta \varphi_{e}$, between the two signals is calculated by using the arc tangent function:(5)$$\Delta \varphi_{e}=\arctan\left(\frac{Q_{P}\left(t\right)}{I_{P}\left(t\right)}\right).$$

The variance of the phase difference can be expressed as
(6)$$\mathrm{var}\left(\Delta \varphi_{e}\right)=\frac{B}{C/N_{0}}\left(1+\frac{1}{2T_{coh}C/N_{0}}\right)rad^{2},$$
where *B* is the noise bandwidth, $T_{coh}$ is the integration time, and $C/N_{0}$ is the carrier-to-noise ratio information of BeiDou signals [31], which can be expressed as
(7)$$C/N_{0}=\frac{1}{T_{coh}}\left(\frac{M}{M+2}Z-1\right).$$

In this paper, when calculating the carrier-to-noise ratio of BeiDou reflected signals, the *M* value is 20, and the *Z* value can be expressed as
(8)$$Z=\frac{\sum_{n=1}^{m}\left({I_{P}}^{2}\left(n\right)+{Q_{P}}^{2}\left(n\right)\right)}{2\sum_{n=1}^{m}{I_{N}}^{2}\left(n\right)},$$
where $I_{P}$ and $Q_{P}$ represent the results of coherent integration, and $I_{N}$ represents the results of the coherent integration of noise. When there is a ground target in the footprint of the antenna, the geometrical relationship is as shown in Figure 3. The received reflected signal can be expressed as
(9)$${u_{i}}^{*}=u_{{}_{ref-ground}}^{*}\left(t\right)+u_{{}_{ref-target}}^{*}\left(t\right)-\xi_{1}-\xi_{2},$$
where $u_{{}_{ref-ground}}^{*}\left(t\right)$ is the reflected signal from the ground, which does not include the reflected signal from the target itself, which is represented by $u_{{}_{ref-target}}^{*}\left(t\right)$. $\xi_{1}$ is the signal reflected twice between the ground and the target, which is represented by $r_{s}$ in Figure 3. $\xi_{2}$ is a part of the lost signal in Figure 3, which is shown in the blue part.

The phase difference information of the reflected signal in the presence of the target can be expressed as
(10)$${\Delta \varphi_{e}}^{*}=\arctan\left(\frac{{Q^{*}}_{P\_ground}\left(t\right)+{Q^{*}}_{P\_target}\left(t\right)+{Q^{*}}_{P\_\xi_{1}}\left(t\right)+{Q^{*}}_{P\_\xi_{2}}\left(t\right)}{{I^{*}}_{P\_ground}\left(t\right)+{I^{*}}_{P\_target}\left(t\right)+{I^{*}}_{P\_\xi_{1}}\left(t\right)+{I^{*}}_{P\_\xi_{2}}\left(t\right)}\right).$$

Because of the uncertainty of $\xi_{1}$ and $\xi_{2}$, the phase difference information cannot be calculated accurately, and the stable state cannot be achieved. In order to express the existence and non-existence of a target in a unified way, the phase difference information can also be expressed as
(11)$$\Delta \varphi_{e}=\arctan\left(\frac{\sin(\Delta \varphi_{e})}{\beta^{-1}+\cos\left(\Delta \varphi_{e}\right)}\right),$$
where β is the signal attenuation coefficient. When a target appears, the stability of the tracking loop is affected, and the phase difference between the local replica and the received signal is close to 90${}^{{}^{\circ}}$. When $\beta^{-1}$ is close to 0, it means that there is no target, and the phase difference information can be calculated normally.

Assuming that the carrier-to-noise ratio of the reflected signal is represented as ${\left(C/N_{0}\right)}_{target}$, when there is no target, it is ${\left(C/N_{0}\right)}_{ground}$. If the target is located in the first Fresnel diffraction region of the satellite when the reflector antenna is in the azimuth of the satellite, the reflected signal from the satellite will decrease because of the existence of the target. In this case, the following relationship between these two parameters can be expressed as
(12)$${\left(C/N_{0}\right)}_{ground}>{\left(C/N_{0}\right)}_{target}.$$

When the direction of the reflector antenna deviates from the azimuth of the satellite, the change in the target position complicates the reflection of the satellite signal and changes the strength of the parameters in Equation (Equation 9). In this case, the power of the reflected signal will be enhanced at a certain time when the target is moving. Additionally, the following relationship between these two parameters can be expressed:(13)$${\left(C/N_{0}\right)}_{ground}<{\left(C/N_{0}\right)}_{target}.$$

Here, Equations (12) and (13) describe changes in the intensity of the reflected signal when the target moves, which are verified in the second experiment presented in this paper.

### 2.2. Analysis of Detection Probability

Based on the target detection mechanism, an equivalence analysis of whether the target appears or not is carried out. Additionally, a binary construction model is established (seen in Figure 4), which can be expressed as follows:(14)$$H=\left\{\begin{array}{c} H_{0}:\;\;\;\;s\left(t\right)=s_{0}\left(t\right)+n\;,\;\;\;\;\;\;\;\;\;\;\;\;\;\;\;\;\;\mathrm{no}\;\;\;\;\mathrm{target}\;\;\;\;\;\;\;\;\;\;\;\;\;\;\;\;\;\;\;\;\;\;\;\;\;\;\;\;\;\;\;\;\\ H_{1}:\;\;\;\;s\left(t\right)=s_{1}\left(t\right)+n\;,\;\;\;\;\;\;\;\;\;\;\;\;\;\;\;\;\;\;\;\mathrm{target} \end{array}\right.$$
where $s_{0}\left(t\right)$ denotes the signal received by the LHCP antenna when the target does not exist, $s_{1}\left(t\right)$ denotes the signal received when the target exists, and the observation noise *n* is Gaussian noise with a mean value of 0 and a variance of $\sigma^{2}$. When the signal s(t) is observed N times in the range of 0 to *T*, its two probability distributions can be expressed as follows:(15)$$\begin{array}{cc} p\left(S\left|H_{0}\right.\right) & =p\left(s_{1}s_{2}\cdots s_{N}/H_{0}\right)\\ & =p\left(s_{1}/H_{0}\right)\cdot p\left(s_{2}/H_{0}\right)\cdots p\left(s_{N}/H_{0}\right)\; \end{array}$$
and
(16)$$\begin{array}{cc} p\left(S\left|H_{1}\right.\right) & =p\left(s_{1}s_{2}\cdots s_{N}/H_{1}\right)\\ & =p\left(s_{1}/H_{1}\right)\cdot p\left(s_{2}/H_{1}\right)\cdots p\left(s_{N}/H_{1}\right)\;. \end{array}$$

Among them, the probability density of each observation can be expressed as
(17)$$p\left(s_{k}\left|H_{0}\right.\right)=\frac{1}{\sqrt{2\pi }\sigma }\exp\left(-\frac{{\left(s_{k}-s_{0k}\right)}^{2}}{2\sigma^{2}}\right)$$
and
(18)$$p\left(s_{k}\left|H_{1}\right.\right)=\frac{1}{\sqrt{2\pi }\sigma }\exp\left(-\frac{{\left(s_{k}-s_{1k}\right)}^{2}}{2\sigma^{2}}\right).$$

The likelihood function ratio of the binary hypothesis model can be expressed as
(19)$$l\left(s\left(t\right)\right)=\frac{P(s\left(t\right)\left|H_{1}\right.)}{P(s\left(t\right)\left|H_{0}\right.)}\begin{array}{c} \overset{H_{1}}{>}\\ \underset{H_{0}}{<} \end{array}\frac{\left(C_{10}-C_{00}\right)}{\left(C_{01}-C_{11}\right)}\frac{P(H_{0})}{P(H_{1})}=l_{0}$$
where $l_{0}$ is the threshold value. $C_{00}$, $C_{01}$, $C_{10}$, and $C_{11}$ are the cost factors, which are determined by the criterion of selection. In the optimal communication system, the minimum total error probability criterion (namely Bayesian criterion) is usually used, where the cost factors $C_{00}=C_{00}=0$ and $C_{10}=C_{01}=1$.

As shown in Figure 5, the results of the target detection can be divided into four situations: (1) $P(H_{0}/H_{0})$, which means there is no target and the detection result is correct; (2) $P(H_{0}/H_{1})$, which means there is a target and the detection result is wrong; (3) $P(H_{1}/H_{0})$, which means there is no target and the detection result is wrong; and (4) $P(H_{1}/H_{1})$, which means there is a target and the detection result is correct.

Based on this, the false alarm probability $P_{f}$, the detection probability $P_{d}$, and the miss probability $P_{m}$ of the target detection can be expressed as
(20)$$\begin{array}{cc} P_{f} & =P\left(H_{1}/H_{0}\right)=\int_{l_{0}}^{\infty }P(s\left(t\right)|H_{0})dt\\ & =\int_{l_{0}}^{\infty }\frac{1}{\sqrt{2\pi }\sigma }\exp\left(-\frac{{\left(s\left(t\right)-s_{0}\left(t\right)\right)}^{2}}{2\sigma^{2}}\right)dt \end{array}$$
and
(21)$$\begin{array}{cc} P_{m} & =P\left(H_{0}/H_{1}\right)=\int_{-\infty }^{l_{0}}P(s\left(t\right)|H_{1})dt\\ & =\int_{-\infty }^{l_{0}}\frac{1}{\sqrt{2\pi }\sigma }\exp\left(-\frac{{\left(s\left(t\right)-s_{1}\left(t\right)\right)}^{2}}{2\sigma^{2}}\right)dt \end{array}$$
and
(22)$$\begin{array}{cc} P_{D} & =P\left(H_{1}/H_{1}\right)=\int_{l_{0}}^{\infty }P(s\left(t\right)|H_{1})dt\\ & =\int_{l_{0}}^{\infty }\frac{1}{\sqrt{2\pi }\sigma }\exp\left(-\frac{{\left(s\left(t\right)-s_{1}\left(t\right)\right)}^{2}}{2\sigma^{2}}\right)dt. \end{array}$$

## 3. Experimental Campaign

### 3.1. Description of the First Experiment

The experiment using the BeiDou reflected signal to detect ground targets has higher requirements for the surrounding environment. Because this experiment is a preliminary verification of the technology, if the surrounding environment is complex, the reflected signal received by the antenna will have interference, which will affect the analysis of the results. Based on this, the selected experimental site is located in the suburban area of Beijing, whose latitude and longitude are $39.69^{{}^{\circ}}$ and $116.69^{{}^{\circ}}$, respectively. There are no tall buildings around it, which avoids the influence of surrounding environment on the BeiDou signal.

A LHCP antenna with a gain of 12 dB was mounted on a triangular frame with a height of 1.4 m. The antenna was placed nearly downward horizontally, whose direction was west (seen in Figure 6), and it was able to receive the GPS L1 and BeiDou B1 signals at the same time. In this experiment, only the reflection signal of BeiDou B1 was collected. At the same time, a 3 dB omnidirectional Right-Handed Circularly Polarized (RHCP) antenna was placed vertically on the top of the triangle. It should be noted that, in this experiment, the function of the RHCP antenna was just to acquire the signal quality of the BeiDou satellite in real-time. As shown in Figure 4, the BeiDou reflected signal was received by a dual-channel collector and stored on a computer hard disk in real-time. The sampling frequency of the dual-channel collector was 16.368 MHz, and the digital intermediate frequency was 4.992 MHz. This was previously successfully applied in the typhoon experiment at the Yangjiang site [22]. In addition, there were about 8 M bytes of buffer data in the collector, which needed to be skipped during data processing.

In this experiment, metal boxes were selected as the target, and the size of box is shown in Table 1. In order to verify the feasibility of using the BeiDou B1 frequency signal for ground target detection, a total of 70 s of data were collected, and the target was placed in the footprint of the LHCP antenna for two time periods (10–20 s and 40–55 s). For other periods, the metal box was removed. This allowed the difference between the two cases to be compared. It should be noted that the target placement time started from the time of signal acquisition.

### 3.2. Description of the Second Experiment

In the first experiment, the fact that the target was put in and out by a human would have impacted the detection results. At the same time, the target position was fixed and static, so it was not easy to observe the influence of other satellite reflection signals. In order to solve this problem, a second experiment was carried out in this paper. In this experiment, the signal acquisition equipment, antenna erection height, and the experimental site were all kept unchanged, and the experimental time was 2019_04_04_15_52_13. The experimental setup is shown in Figure 7.

It should be pointed out that the LHCP antenna was oriented to the south, and the tripod was placed in the middle of the road surface. The target moved between AB, with a width of 6 m. Point C indicates that the target moved directly below the antenna. The target was towed by two ropes, moving uniformly in the east and west directions, as shown in Figure 8.

In Figure 8, the *x*-axis was west-orientated, and the *y*-axis was south-oriented. EFG is the position of the tripod, and the blue quadrilateral area ABCD is the moving range of the target. The vertical distance between CD and EF was 0.5 m, and the distance of AB was 6 m.

There were some buffer data when the dual-channel collector collected BeiDou signals. Therefore, in the experiment, the target moved from position A in about 10 s after data acquisition, and the data acquisition time was about 3 min. At the same time, the stopwatch was used to record the specific time when the target passed directly below the antenna. The main information is shown in Table 2. Based on this, the moving speed of the target was calculated to be about 0.5 m/s.

## 4. Experimental Results

### 4.1. Results of the First Experiment

The BeiDou software receiver was used to process the collected BeiDou reflected signals. Three satellites were captured and tracked, which were PRN9, PRN16, and PRN24 respectively. Among them, PRN9 is an IGSO satellite, while PRN16 and PRN24 are MEO satellites. BeiDou IGSO satellites have the same orbital altitude as the GEO satellite, so they have the same orbital period as the Earth’s rotation period. Because the orbital inclination of the BeiDou PRN9 satellite is 54.583${}^{{}^{\circ}}$, its sub-satellite point trajectory has an “8” shape distribution with the equator (about 120${}^{{}^{\circ}}$ east longitude) as the symmetrical axis on the ground, which can cover most of the area of China [32].

In particular, due to the constellation characteristics of the BeiDou IGSO satellite, its visible time in China is much longer than the other MEO satellites. In this paper, the antenna was facing west to verify the reliability of the BeiDou IGSO satellite in the field of target detection. It should be noted that if we want to monitor whether there are targets in a fixed area for a long time, we need to adjust the direction of the antenna and choose BeiDou GEO satellite as the signal source. This is because, in the application of reflected signals, the GEO satellite can maintain a stable geometric relationship.

According to the longitude and latitude of the experimental site, the BeiDou satellite skyplot at the time of data acquisition, 2018_11_15_15_35_32, can be obtained as shown in Figure 9. Based on this, the azimuth and altitude information of the three BeiDou satellites can be extracted, as shown in Table 3.

In the remote sensing field, when using GNSS reflected signals, the peak power of the received signal is typically reflected from the surface surrounding the specular point, where the incident and reflected angles are equal [33]. In addition, the first glistening zone surrounding the specular point contributes most of the power of the reflected signal. According to the method presented in [34], the specular points on the ground surface between the three BeiDou satellites and the LHCP antenna can be calculated, the relationship of which is shown in Figure 10. Blue represents PRN16, red represents PRN9, and green represents PRN 24. The ellipses depict the first Fresnel zone, and the pots, numbered 2, 3, and 4, depict the specular reflection point. The bold dark ellipse is the LHCP antenna’s footprint. The bold dark pot stands for receiver number 1, considering the origin of the reference system. The blue rectangle represents the mental box.

As can be seen from Figure 10, the target was located nearest to the specular point of the PRN9 satellite and farther away from the specular points of PRN24 and PRN16 satellites. By tracking the reflected signals of the three satellites, the phase difference information in the tracking loop was obtained, as shown in Figure 11a, Figure 12a and Figure 13a. In the tracking loop, the output interval of the phase difference information was 1 ms. In the experiment, the time units of the box placement were seconds. In order to achieve the same time units, the maximum value of each 1000 phase difference information values was selected, as shown in Figure 11b, Figure 12b and Figure 13b.

As can be seen from Figure 11, there were two distinct changes in the phase difference information in the PLL loop when tracking the PRN9 satellite reflected signal. When the target appeared in the coverage of the antenna, the phase difference information of the reflected signal was kept near 90${}^{{}^{\circ}}$, and the duration of the phenomenon depended on the existence time of the target. If the target was removed, the phase difference value would return back to the normal level—near 20${}^{{}^{\circ}}$. After selecting the maximum value, the two periods of significant changes were obtained—about 12–22 s and 41–56 s. In the experiment, the target placement times were 10–20 s and 40–55 s.

Compared with the processing results, there was a deviation of 1–2 s. There are two main reasons for this phenomenon. Firstly, there were some errors in the time of target placement and removal. Secondly, there were buffer data in the dual channel collector when receiving BeiDou signals. In order to avoid its impact, it is necessary to skip this data segment in the process of calculating the phase difference information of the BeiDou reflected signal. The number of data bytes skipped in the software receiver was $4\times 10^{7}$. However, the results are very consistent with the actual situation when the duration time of phase difference information change is observed.

The phase difference information of PRN16 showed no obvious change, which can be seen in Figure 12. Except for the abnormal protrusions near 11 s, most of the phase difference values were in a stable range. The analysis results of PRN24 from Figure 13 show that the phase difference information increased abruptly to 90${}^{{}^{\circ}}$ four times, near 10, 22, 41, and 58 s, respectively, and then it returned to its normal value; this process lasted for about 1 to 2 s, while the phase difference of other time periods remained in a small range. These moments were close to the target’s placement time, but there was no continuous time period. This phenomenon was caused by the influence of the signal from the placement of the metal boxes. By comparing and analyzing the results of the phase differences of the three BeiDou satellites, it was concluded that the position relationship between the target and the specular points determines whether the ground target can be detected by the reflected signal or not.

It is well known that the specular point determines the location of the first Fenier reflection region [35,36]. This phenomenon shows that when the location of the target is not only in the coverage area of the LHCP antenna but also in the first Fresnel area of BeiDou satellites, it means that the LHCP antenna can receive the reflected signal from the target surface. In this case, the phase difference information of the reflected signal will be greatly affected, and the time of target placement can be calculated by its change in duration, as demonstrated by the results obtained from PRN9 in the experiment. On the contrary, if the location of the target does not satisfy this relationship, as occurred for BeiDou PRN16 and PRN24, it will not be detected by the method mentioned.

By calculating the carrier-to-noise ratio of the three satellites [37,38], the variation trend of the PRN9 satellite was found to be more obvious than that of the other two satellites, which is consistent with the time when the target appeared (see in Figure 14). Compared with the situation when the target did not appear, the CNR of the BeiDou reflected signal decreased by about 7 dB. Through the probability density analysis of the phase difference information and the calculation of CNR information, the feasibility of using phase difference information to detect ground targets was verified. This also verifies the existence of the reflected signal variation described in Formulas (13) and (14). However, there were some problems in the validation experiment, and thus it could not fully explain the target detection using the BeiDou reflection signals.

### 4.2. Results of the Second Experiment

The results of the first experiment suggest that the target information can be clearly seen by the reflection signals of the PRN9 satellite. The reflection signals of PRN16 and PRN24 satellites showed no obvious changes. However, due to the design of the experiment, there were several problems, as follows: (1) the target was put in and out by a human, which would have had some influence on the detection results (as shown in Figure 13); (2) because the target was placed in the same position each time, the impact of the target on the PRN16 and PRN24 satellite reflection signals could not be accurately judged, and the effective range of target detection could not be judged by the first experiment; (3) the CNR information showed obvious changes in the PRN9 satellite which coincided with the target placement time, but PRN24 increased. Through the results of the first experiment, this phenomenon cannot be explained accurately.

In order to solve these problems, the second experiment was designed to build on the first experiment; the specific experimental information is detailed in Section 3. The location information and data acquisition time of the experimental site were used to obtain the BeiDou satellite sky map, as shown in Figure 15. The direction indicated by the black arrow is the direction of the reflection antenna. Five satellites, PRN2, PRN3, PRN8, PRN23 and PRN27, were captured and tracked by processing the collected BeiDou satellite reflection signals. The state information of the five satellites was obtained with the satellite sky map, as shown in Table 4.

The impact of the target on the reflected signal was analyzed in combination with the CNR information of the five satellite reflection signals (as shown in Figure 16). To process the phase difference information of the BeiDou reflection signal, an analysis of I and Q data for each satellite was carried out in the second experiment. The concrete results are shown in Figure 17, Figure 18, Figure 19, Figure 20 and Figure 21.

The I branch component data of the PRN2 satellite changed very little; amplitude changes were seen, but the phenomenon was not obvious. As shown in Figure 17d, the change trend of the phase difference had several protrusions, but the change range was not large, and there was no regularity, so it was difficult to find a reasonable judgment threshold. The phase bulge time was very different from the time when the target passes under the antenna, and the overall judgment effect was poor. Combined with the carrier-to-noise ratio information in Figure 16a, the signal-to-noise ratio information of the satellite increased or decreased several times at the time that the target appeared, which indicates that the target has a certain influence on the reflection signal of the satellite.

A shown in Figure 18, the change trend of phase difference of PRN3 was close to that of PRN2, and the appearance of the target cannot be accurately achieved by phase difference information alone. However, there was a distinct difference between the PRN2 and PRN3 satellites. The I branch component data of the PRN3 satellite changed obviously and increased obviously. Moreover, the time when the magnitude increases coincided with the time when the target appeared. From Figure 16b, we can see that the CNR information of PRN3 satellite increased at 14 time nodes, which indicates that the intensity of the reflected signal from the PRN3 satellite increased when the target appeared.

It can see that there were 14 obvious signal changes in the I branch component data of the PRN8 satellite from Figure 19. Unlike the PRN3, PRN23 and PRN27 satellites, the I branch component of the PRN8 satellite changed in a “depression” state. The changing trend of the phase difference shows that the change was consistent with the number of times the target passes through, and the time duration of each change was 2 s. It should be mentioned that the change in the PRN8 phase difference was the same as that of the PRN9 satellite in the first experiment. At the same time, combined with the carrier-to-noise ratio information of PRN8 shown in Figure 16e, we can see that there were 14 notches in the carrier-to-noise ratio, which indicates that the reflected signal intensity from PRN8 satellite decreased when the target passed under the antenna.

In Figure 20, the overall change of PRN23 satellite was similar to that of PRN3 satellite. A slight difference is that there were 14 increased data segments in the I branch component data, but the amplitude was not large, around 3000, while the increase in the I branch component data of the PRN3 satellite was maintained at 4000. Compared with the carrier-to-noise ratio information of the two satellites, the signal intensity of the PRN23 satellite was also significantly less than that of the PRN3 satellite. At the same time, the carrier-to-noise ratio of the PRN23 satellite fluctuated upwards and downwards with the movement of the target, and there was no indication that the CNR information of the PRN3 satellite increased regularly, as shown in Figure 16c.

As shown in Figure 21, the changes in the I branch component data and phase difference information of the PRN27 satellite were quite different from those of the other satellites. As shown in Figure 21a, the I branch component changed 14 times. Unlike several other satellites, the amplitude of the I branch component data increased and decreased simultaneously when the target passed under the antenna. Combined with the CNR information shown in Figure 16d, the same phenomenon appeared in the PRN27 satellite reflection signal intensity. In addition, the change in the phase difference of the target was also reflected (as shown in Figure 16d), but the change range was small, within $-50^{{}^{\circ}}$ to 50${}^{{}^{\circ}}$, while that of the PRN8 satellite was near 90${}^{{}^{\circ}}$. In order to explain the special phenomenon of five satellite reflected signals more clearly, a period of data was intercepted from the I branch component of the five satellites; the time length was 6000–28,000 ms. This time period included the first time the target moved (from west to east) and the second time it moved (from east to west) under the antenna, as shown in Figure 22.

Green box 1 in Figure 22e clearly shows that when the target passed under the antenna for the first time (from west to east), the I branch component of the PRN27 satellite underwent a process of oscillation, descent, and ascent, and recovered to its normal level after the target passed through. Green box 2 shows the change in the I branch component when the target passed under the antenna for the second time (from east to west), which was contrary to the trend of box 1. Comparing the I branch component data in the green box in Figure 22a, we also observed the same change, but the change range was small. It should be mentioned that the green box in Figure 22d indicates the change trend of the PRN23 satellite I branch data during the second time that the target passed under the antenna (from west to east), similar to that of PRN27 satellite green box 2 in Figure 22e. However, there were two differences: (1) the moving direction of the target was different when it passed through the antenna, and (2) the range of variation for the I branch component data of the PRN23 satellite was smaller than that of the latter. The red dotted line in Figure 22e is the time period during which the I branch component declined. It is the reason why the phase difference of the PRN27 satellite (shown in Figure 21d) changed significantly. However, the decrease in PRN8 was not as large as that of the red dotted line of PRN8 in Figure 22c, so the change in the phase difference of PRN27 was smaller than that of PRN8.

In the second experiment, many new phenomena were found through the above analysis because of the long data acquisition period and the avoidance of human influence. Different BeiDou satellites were shown to detect 14 round-trip motions of targets, but the detection parameters and performance were different. The target detection events obtained by different parameters, such as I branch component data, phase difference information, and carrier-to-noise ratio information, are shown in Table 5. It should be pointed out that the time results for PRN2 and PRN23 were estimated based on the I branch component data from Figure 17a and Figure 20a. In order to unify the time units, the values were rounded.

It can be seen from Table 5 that the PRN8 satellite had the best target detection performance. It detected targets through I branch component data, phase difference information, and carrier-to-noise ratio information, and the trends of the three parameters were consistent with those of the PRN9 satellite in the first experiment. Comparing the time recorded when the target passed under the antenna, it was found that there was an error of about two seconds between the result of PRN8 inversion and that of PRN8 inversion. This is a normal phenomenon that occurs because data processing needs to skip the buffer data section during acquisition. The PRN3 satellite found obvious changes mainly through I branch component data and carrier-to-noise ratio information, which were basically consistent with the detection results of the PRN8 satellite, but the trends of the I branch component data and carrier-to-noise ratio were different between the two satellites. Comparing and observing the detection results of PRN2 and PRN8, when the target moved from east to west (that is, when the number of times the target passed under the antenna was odd), the detection time was basically the same. When the target moved from west to east (that is, the number of times the target passed under the antenna was even), the detection time of PRN2 satellite is about 2 s earlier than that of PRN8 satellite. The PRN27 showed the same changes, but the difference was that when the target moved from west to east, the detection time of PRN27 was about 1 s ahead of the PRN8 satellite. Comparing the detection results of the PRN23 and PRN8 satellites, when the target moved from east to west, the detection time of the PRN23 satellite was about 2 s earlier than that of the PRN8 satellite. When the target moved from west to east, the detection time was the same.

In order to explain the above phenomena effectively, the positions of specular points between the five satellites and the antennas were calculated by combining the position information for satellites and antennas, and a geometric diagram was drawn, as shown in Figure 23.

According to the geometric relationship shown in Figure 23, the processing results of the reflected signals of five BeiDou satellites can be explained. When the target moved to different positions, the reflected signal of the target changed greatly. When the target was in the first Fresnel diffraction region of the satellite, the reflection signal of the target decreased, and the reduction time was consistent with the time when the target passed directly below the antenna, as shown in the CNR information of the PRN8 satellite. When the target was not in the first Fresnel diffraction region, but still in the second or third Fresnel diffraction region, the reflected signals from other satellites were effectively received by the antenna, but the intensity of the reflected signals changed significantly and did not appear to show the same situation as the PRN8 satellite, as shown by the information from PRN23, PRN2, PRN27, and PRN3.

From Table 5, each satellite detects the target at different times. Combining with the position relationship in Figure 23, it can be found that this was closely related to the position of the specular point. According to the information shown in Table 5, when the target moved from east to west, the satellites detecting the target were PRN23, PRN8 (PRN3), PRN27 and PRN2 in turn, and the location of the mirror points is the same from east to west. It should be mentioned that PRN23 and PRN2 were about 2 s behind PRN8 satellite when they detected the target, which shows that the effective detection range of the antenna was about 1 m below the antenna (the average velocity of the target motion is 0.5 m/s). It should be mentioned that the scope was based on the span of the target in the east and west.

## 5. Conclusions

In this paper, the mechanism of detecting targets using BeiDou reflection signals was discussed, and validation tests were carried out using moving and stationary targets. The detection performance of targets was analyzed by parameters such as the I branch component data, phase difference information, and carrier-to-noise ratio information.

By comparing the results from the two experiments, the following conclusions can be drawn: 1. When the target is completely in the first Fenier diffraction zone and in the footprint of the reflector antenna, the carrier-to-noise ratio of the BeiDou satellite reflector signal decreases, and it can be detected through the I branch component data and carrier-to-noise ratio information (as shown for the PRN9 satellite in the first experiment and the PRN8 satellite in the second experiment); 2. When the target moves to different positions, the carrier-to-noise ratio of the BeiDou satellite reflection signal does not decrease due to the relative change between the positions of the target and the mirror point. When the mirror position does not appear directly below the antenna but is offset, and the carrier-to-noise ratio information of the BeiDou satellite reflection signal increases, the target can be effectively detected by using I branch compomemt data and carrier-to-noise ratio information, but there is no obvious change in phase difference information, as shown for the results of the PRN3 satellite in the second experiment. When the mirror offset distance increases, the target may appear in the second or third diffraction region of the satellite, and the reflected signal from the target can still be received. When the target moves, the change in the reflected signal is more complex than the two cases mentioned above, and a decrease and increase of the reflected signal occurs. The order of the reflected signal is related to the direction of the target motion and the position of the mirror point. In this case, the effective information of the target can be seen through the I branch component data and the carrier-to-noise ratio information; 3. In the second experiment, we can see the range of target detection using the BeiDou satellite reflection signals. Taking the second experiment in this paper as an example, the effective detection range was about 1 m in the antenna position, a range which was not available in the first experiment. 4. For the PRN23, PRN27 and PRN2 satellites, the movement direction of the target can also be related to the changes in the I branch component data.

However, there are still several problems to be solved. First, it is unclear why the intensity of the reflected signal increases when the target appears. Secondly, in the second experiment, the detection range was in the direction of the target’s motion, that is, about 0.5 m before the tripod with the antenna, and the detection range at other distances was not discussed. Thirdly, there has been no detailed theoretical explanation for the relationship between the direction of target motion and the changes in I branch compomemt data. Fourthly, the effective detection thresholds of different detection parameters have not been discussed in detail. In future work, the current author will focus on solving the above problems, improve the theory of using BeiDou satellite reflection signals to detect targets, and improve the practical application ability of the technology.

## Figures and Tables

**Figure 1 sensors-19-02163-f001:**
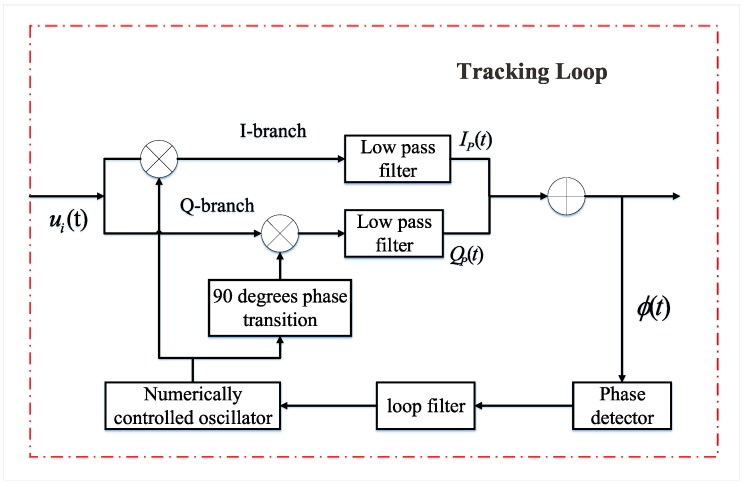
Block diagrams of the phase lock loop in the tracking of BeiDou reflected signals.

**Figure 2 sensors-19-02163-f002:**
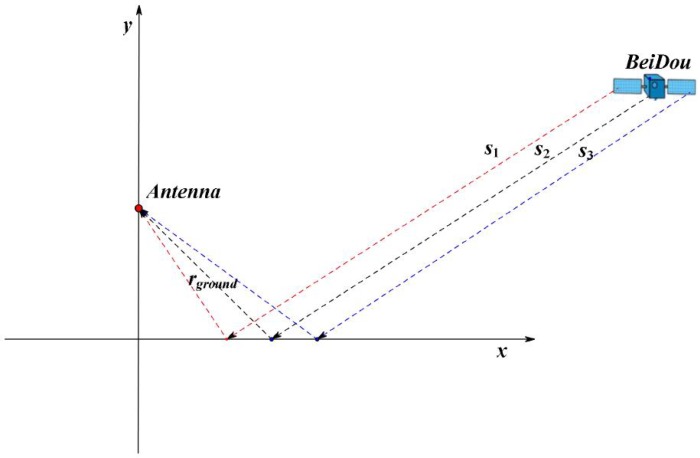
Ground target detection using BeiDou satellite reflected signals when there is no target appearing. $s_{1}$, $s_{2}$, and $s_{3}$ are the direct signals of the BeiDou satellite; $r_{ground}$ is the signal reflected by the ground.

**Figure 3 sensors-19-02163-f003:**
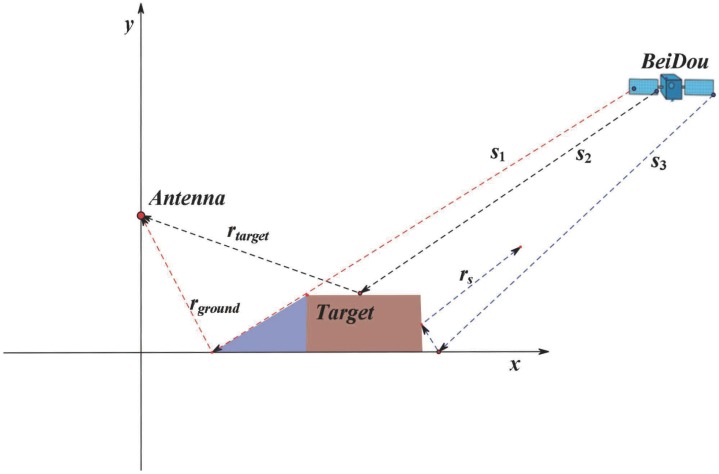
Ground target detection using BeiDou satellite reflected signals. $s_{1}$, $s_{2}$ and $s_{3}$ are the direct signals from the BeiDou satellite; $r_{target}$ is the satellite signal reflected by the target; and $r_{ground}$ is the signal reflected by the ground. The blue part indicates that the BeiDou satellite signal cannot reach the ground surface covered by the target, and $r_{s}$ represents the signal of the BeiDou satellite after secondary reflection between the ground and the target.

**Figure 4 sensors-19-02163-f004:**
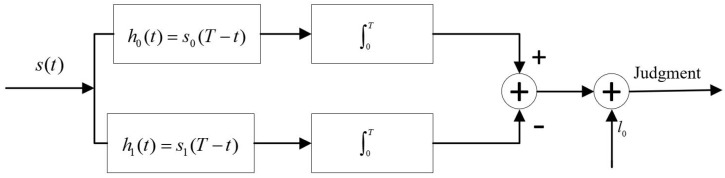
Binary hypothesis model of target detection using BeiDou reflected signals.

**Figure 5 sensors-19-02163-f005:**
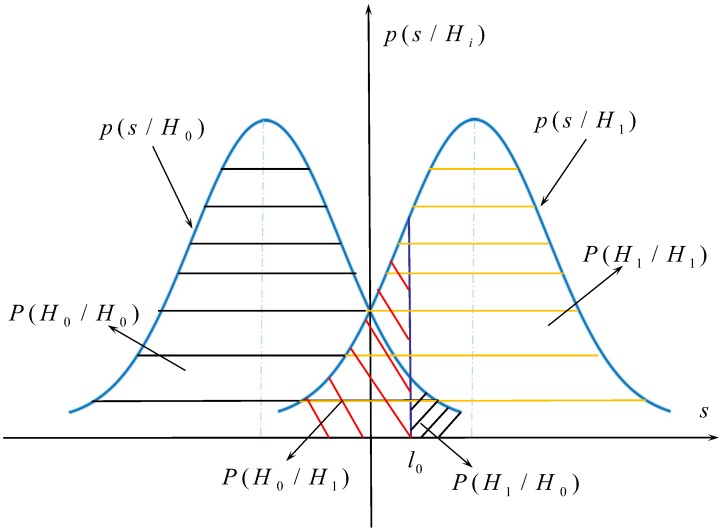
Schematic diagram of the detection probability, false alarm probability, and false alarm probability.

**Figure 6 sensors-19-02163-f006:**
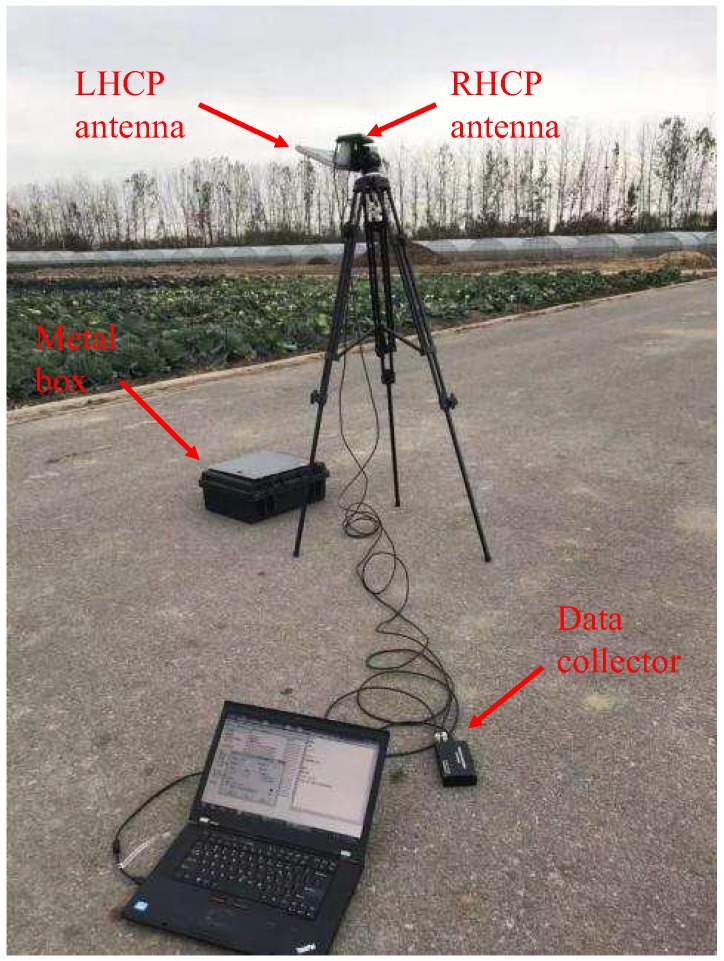
Measurement setup for ground target detection using BeiDou reflected signals. The metal box is placed in the footprint of the left-handed circularly polarized (LHCP) antenna in the experiment.

**Figure 7 sensors-19-02163-f007:**
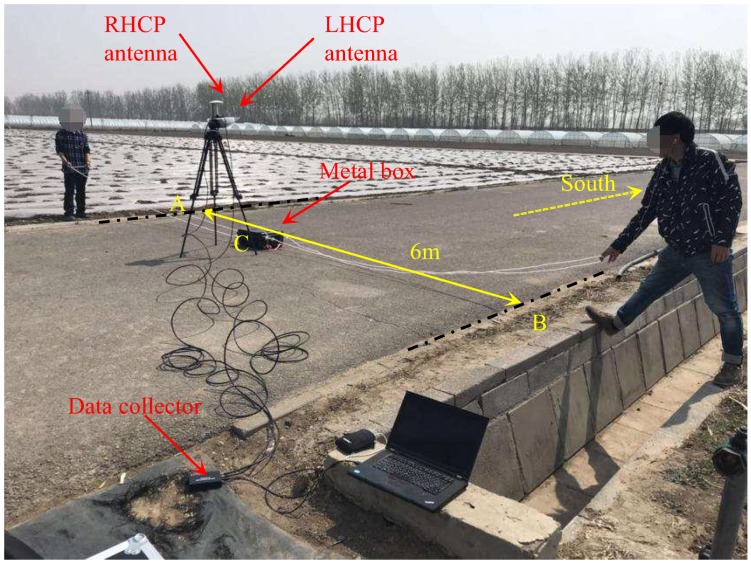
Measurement setup for ground target detection using BeiDou reflected signals in the second experiment.

**Figure 8 sensors-19-02163-f008:**
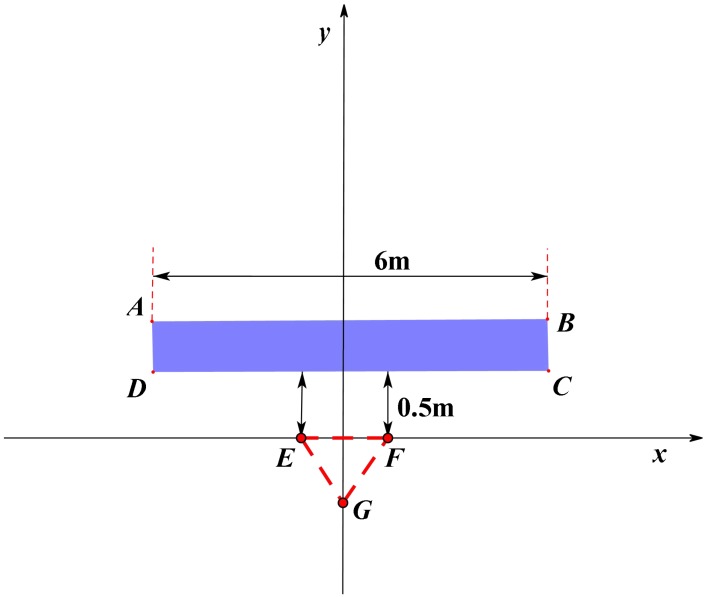
Relationship between the target and the LHCP antenna in the second experiment.

**Figure 9 sensors-19-02163-f009:**
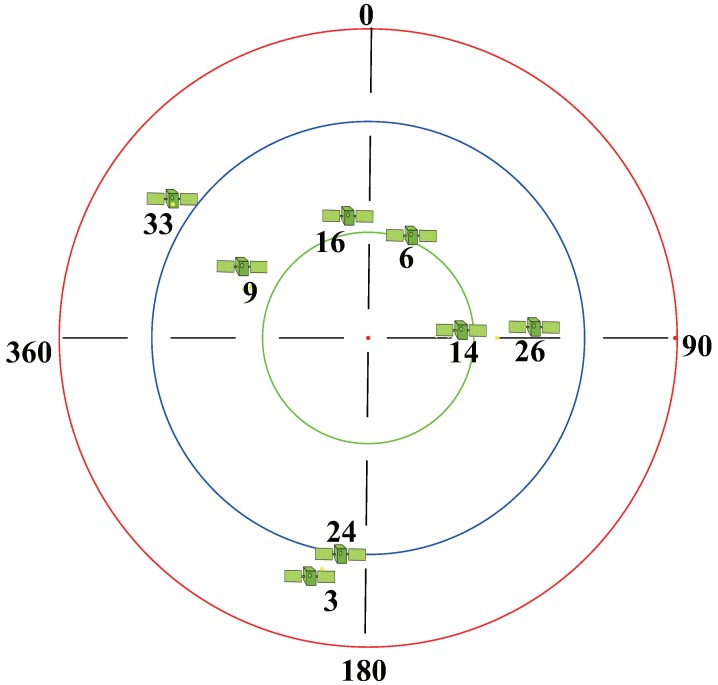
BeiDou satellite skyplot at the data collection time in the first experiment. The latitude and longitude are $39.69^{{}^{\circ}}$ and $116.69^{{}^{\circ}}$.

**Figure 10 sensors-19-02163-f010:**
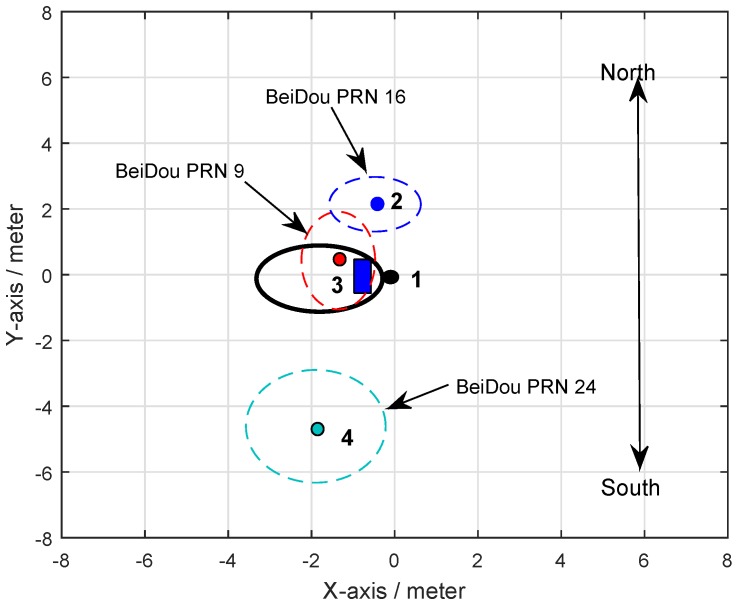
Reflection points of each BeiDou satellite on (*x*, *y*) planes in the first experiment.

**Figure 11 sensors-19-02163-f011:**
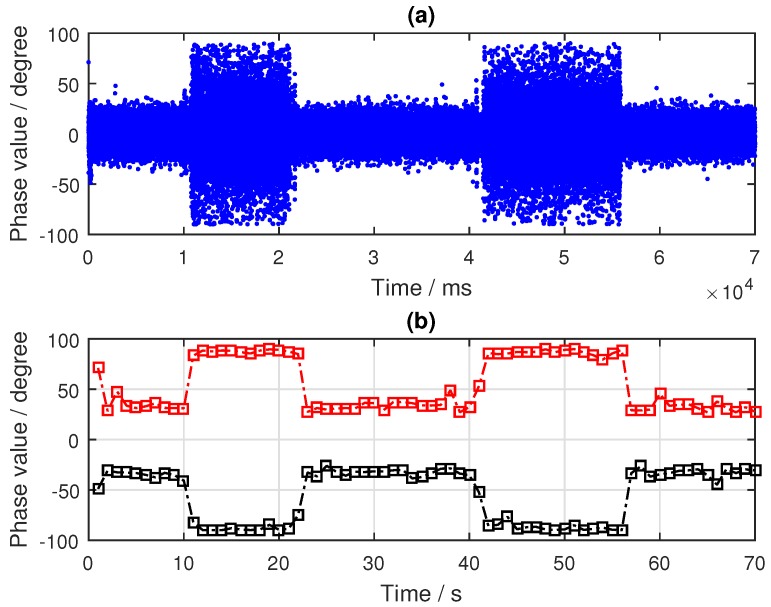
Results of the BeiDou satellite (PRN9) reflected signal: (**a**) phase difference information (in ms); (**b**) phase difference information (in s).

**Figure 12 sensors-19-02163-f012:**
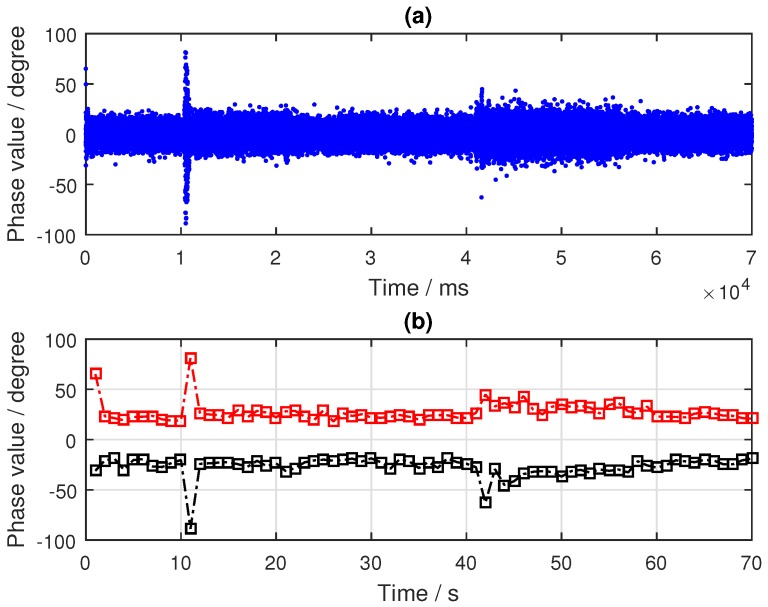
Results of the BeiDou IGSO satellite (PRN16) reflected signal: (**a**) Phase difference information (in ms); (**b**) Phase difference information (in s).

**Figure 13 sensors-19-02163-f013:**
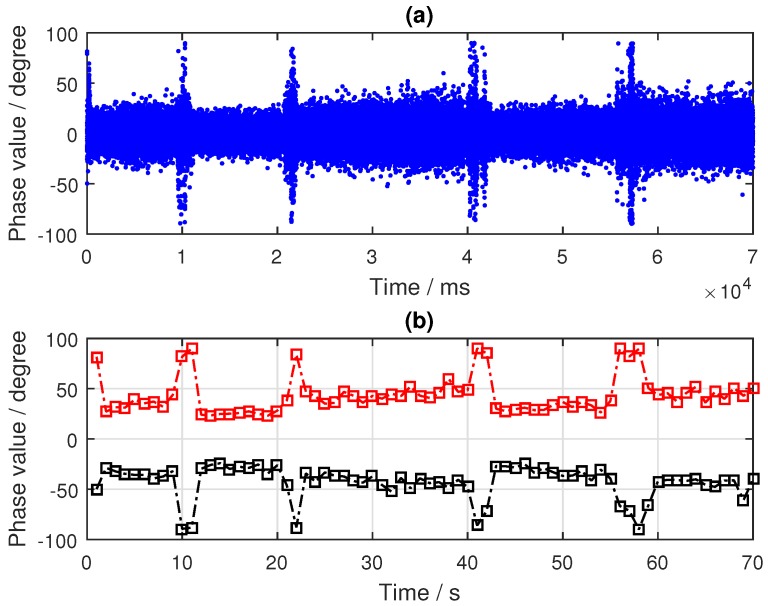
Results of the BeiDou IGSO satellite (PRN24) reflected signal: (**a**) phase difference information (in ms); (**b**) phase difference information (in s).

**Figure 14 sensors-19-02163-f014:**
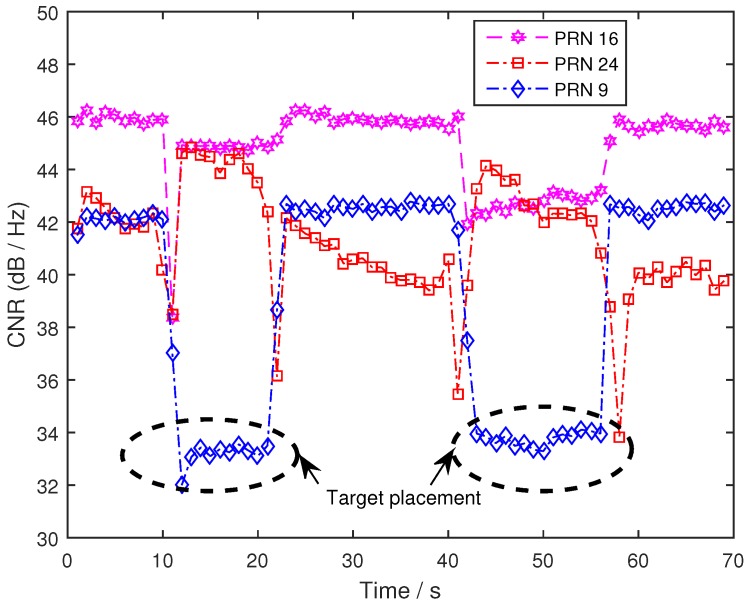
Carrier-to-noise ratio (CNR) information of the three BeiDou reflected signals.

**Figure 15 sensors-19-02163-f015:**
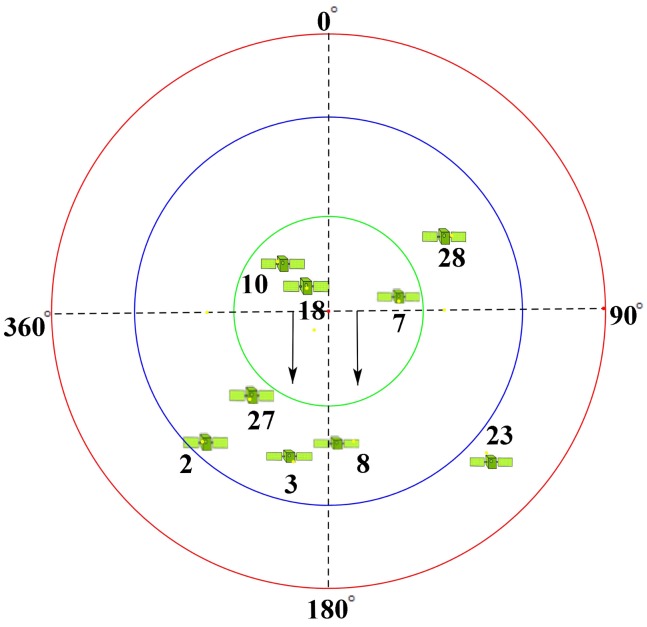
Satellite status information of the BeiDou satellite during data collection.

**Figure 16 sensors-19-02163-f016:**
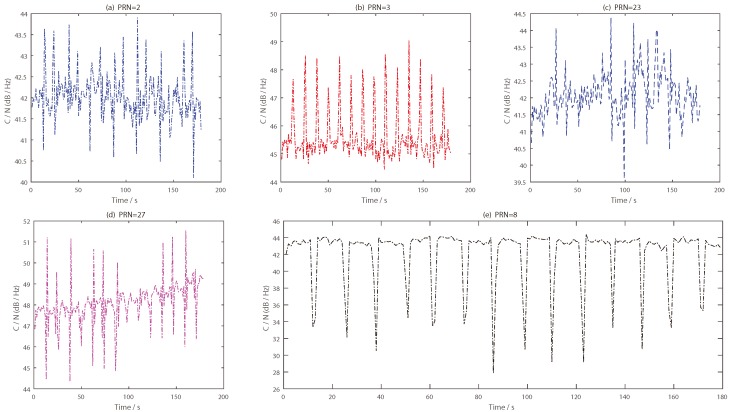
CNR information of five BeiDou satellite reflected signals in the second experiment: (**a**) PRN2 satellite; (**b**) PRN3 satellite; (**c**) PRN23 satellite; (**d**) PRN27 satellite; (**e**) PRN8 satellite.

**Figure 17 sensors-19-02163-f017:**
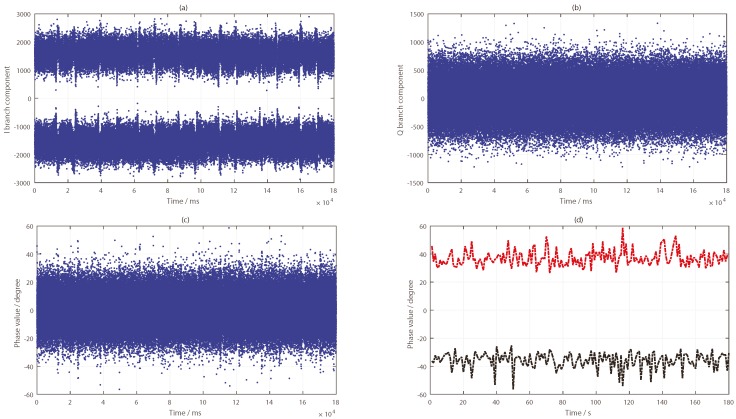
Results for the BeiDou PRN2 satellite reflected signal: (**a**) I branch component data; (**b**) Q branch component data; (**c**) phase difference information (in ms); (**d**) phase difference information (in s).

**Figure 18 sensors-19-02163-f018:**
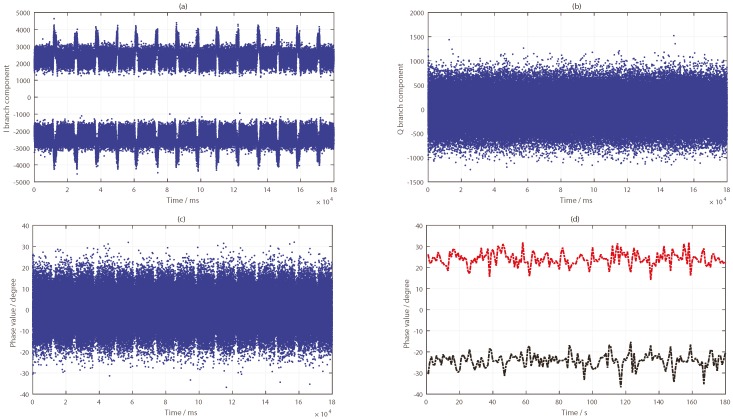
Results for the BeiDou PRN3 satellite reflected signal: (**a**) I branch component data; (**b**) Q branch component data; (**c**) phase difference information (in ms); (**d**) phase difference information (in s).

**Figure 19 sensors-19-02163-f019:**
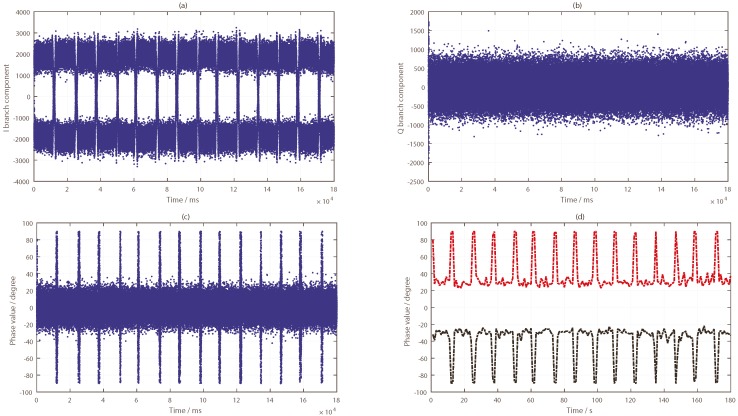
Results of the BeiDou PRN8 satellite reflected signal: (**a**) I branch component data; (**b**) Q branch component data; (**c**) phase difference information (in ms); (**d**) phase difference information (in s).

**Figure 20 sensors-19-02163-f020:**
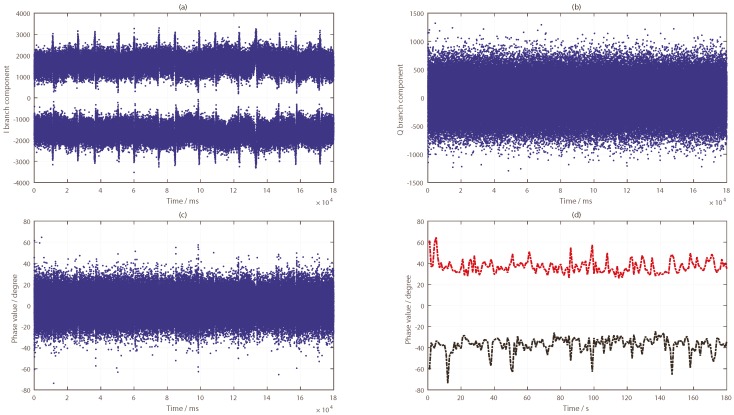
Results of the BeiDou PRN23 satellite reflected signal: (**a**) I branch component data; (**b**) Q branch component data; (**c**) phase difference information (in ms); (**d**). phase difference information (in s).

**Figure 21 sensors-19-02163-f021:**
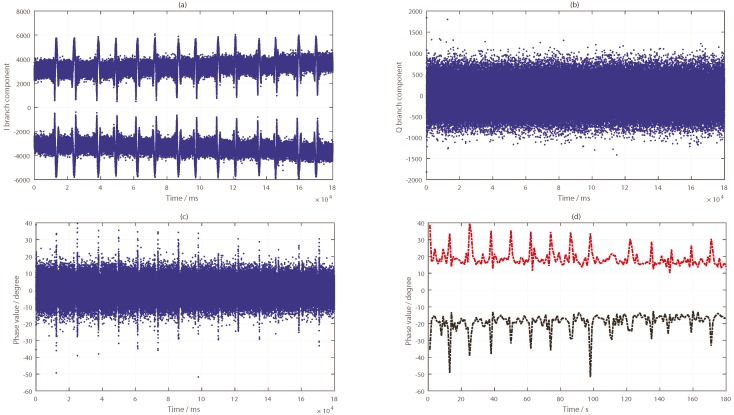
Results of the BeiDou PRN27 satellite reflected signal: (**a**) I branch component data; (**b**) Q branch component data; (**c**) phase difference information (in ms); (**d**). phase difference information (in s).

**Figure 22 sensors-19-02163-f022:**
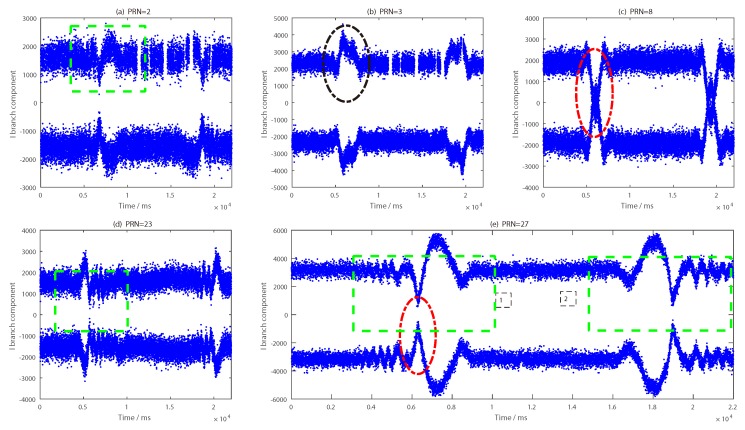
I branch component of five BeiDou satellite reflected signals: (**a**) PRN2 satellite; (**b**) PRN3 satellite; (**c**) PRN8 satellite; (**d**) PRN23 satellite; (**e**) PRN27 satellite.

**Figure 23 sensors-19-02163-f023:**
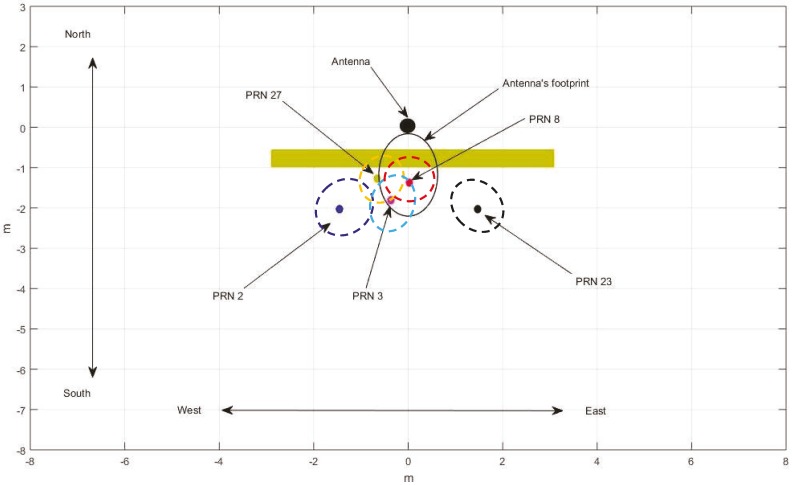
Reflection points of each BeiDou satellite on (*x*, *y*) planes in the second experiment.

**Table 1 sensors-19-02163-t001:** Dimensions of the metal box, which was regarded as the target in the experiment.

Length	Width	Height
40 cm	33 cm	20 cm

**Table 2 sensors-19-02163-t002:** Time information for the target passing through the antenna.

Number of the Target Passing through the Antenna	Time (s)
1	15.05
2	28.08
3	40.09
4	53.05
5	64.05
6	77.04
7	89.02
8	107.07
9	113.04
10	125.09
11	138.02
12	150.00
13	162.00
14	174.07

**Table 3 sensors-19-02163-t003:** Status information of the three acquired BeiDou satellites in the first experiment.

Satellite Orbit	PRN	Elevation Angle	Azimuth
IGSO	9	$61^{{}^{\circ}}$	$304^{{}^{\circ}}$
MEO	16	$70^{{}^{\circ}}$	$348^{{}^{\circ}}$
MEO	24	$45^{{}^{\circ}}$	$189^{{}^{\circ}}$

**Table 4 sensors-19-02163-t004:** Status information of five acquired BeiDou satellites in the second experiment.

Satellite Orbit	PRN	Elevation Angle	Azimuth
GEO	2	$31^{{}^{\circ}}$	$224^{{}^{\circ}}$
GEO	3	$44^{{}^{\circ}}$	$289^{{}^{\circ}}$
IGSO	8	$48^{{}^{\circ}}$	$179^{{}^{\circ}}$
MEO	23	$27^{{}^{\circ}}$	$147^{{}^{\circ}}$
MEO	27	$52^{{}^{\circ}}$	$226^{{}^{\circ}}$

**Table 5 sensors-19-02163-t005:** Target detection time results from the BeiDou satellite reflected signals: (A) I branch component data; (B) phase difference information; (C) carrier-to-noise ratio.

Number	Time (s)	PRN2 (s)	PRN3 (s)	PRN8 (s)	PRN23 (s)	PRN28 (s)
1	15.05	13	13	13	11	13
2	28.08	24	26	26	26	25
3	40.09	38	38	38	36	38
4	53.05	49	50	51	50	50
5	64.05	62	62	62	60	62
6	77.04	73	74	75	75	74
7	89.02	87	86	87	85	86
8	107.07	97	98	99	99	98
9	113.04	111	110	111	109	111
10	125.09	121	123	123	123	122
11	138.02	135	135	135	133	135
12	150.00	145	147	147	147	146
13	162.00	159	159	159	157	159
14	174.07	170	171	172	172	171
Detection parameter	Null	A	A/C	A/B/C	A	A/B

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
