# Peer review of "Performance Analysis of Ground Target Detection Utilizing Beidou Satellite Reflected Signals"

_sensors, 2019, doi:10.3390/s19092163_

Reviewer 1 Report

The paper presents a method about the ground target detection by using reflected signals of BeiDou IGSO satellite. The approach selects the phase difference information as the important observation. While the topic is of some interest to the readers, there are many questions in the paper and they should be addressed. My comments are as follows.

1. Reference [25] has proposed the principle of using phase difference information of GPS reflected signal to detect ground targets. Referring to the literature, what's the contribution of the paper should be clearly descripted.

2. In equation (11), what’s the meaning of parameter \beta?

3. A reasonable explanation is needed for the assumption of equation (12).

4. In equation (18), how to determine the cost factors in general? What factors influence them?

5. Theoretical deduction of the performance and comparisons with other methods are required.

6. In Figure 7, there are eight satellites in the sky. Why just three satellites be captured and tracked?

7. The author concluded the probability density distribution of phase difference can also be used as a criterion for target detection, and the presence of target can also be judged by the CNR information of BeiDou reflected signals. However, which observation is more appropriate?

8. The theoretical analysis of the second part is unclear, for example, the formula 9-11.

9. The experiment is too simple. If the target is always there, what will be the results? And if the goal is big enough to cover the LHCP antenna’s footprint, what will happen?

10. There are many grammar and typo errors. For example, “detetion” in the title “target” in formula 13, etc.

Author Response

Dear reviewers: 

First of all, I am very grateful to the reviewers for their valuable opinions on the first draft of the paper. Using Beidou reflection signal to detect small targets on the ground is a new application direction. Both theoretical research and experimental verification are new explorations of this technology. Because of this, there are also many problems, which need reviewers to provide you with good views and improvements on the problems and shortcomings.

According to the reviewer's opinion, this paper has been revised a lot. In order to better answer the important opinions put forward by the reviewers and to remedy the problems in the first experiment, the author designed and completed the second improvement experiment on April 04, 2019. At the same time, the paper framework and main research contents were rewritten.

Following is the article-by-article response to the reviewer's opinions:

1. Reference [25] has proposed the principle of using phase difference information  of GPS reflected signal to detect ground targets. Referring to the literature, what's  the contribution of the paper should be clearly descripted.

Response:

Thank you very much for this opinion, which is the most important reason for the second experiment.

There are two differences between the preliminary manuscript and literature 25: 1. This paper uses the reflected signal of Beidou IGSO satellite to detect ground targets. 2. The detection probability and false alarm probability of target detection using reflected signal are analyzed.

However, the experiments in the first edition of this paper are basically the same as those in literature 25. Both of them put the target under the antenna in 10-20s and 30-45s, and take it out in other periods.

There is little difference between the two in terms of results or conclusions, and there is no new conclusion in the paper.

Based on it, the second supplementary experiment is carried out. The experiment was conducted on April 4, 2019. The experimental scenario is shown in Figure 1.

Figure 1. Scenario of the second experiment.

In this experiment, the movement of the target is pulled by a rope and moved back and forth in the East-West direction. The total of acquisition time is 3 minutes. In this process, the target passes through the antenna for 14 times.

According to the results of the second experiment, there are several new findings as follows:

1. When the target is moving, the satellite in different azimuth has different time to detect the target, and the detection parameters are also different, which does not depend solely on the phase difference information.

2. The changing trend of I branch component data of individual satellites is closely related to the moving direction of the target, such as the PRN2,PRN 23 and PRN27 in Figure 2.

Figure 2. I branch component of five BeiDou satellite reflected signals: (a) PRN2 satellite; (b) PRN3 satellite; (c) PRN8 satellite; (d) PRN23 satellite; (e) PRN27 satellite.

3. In the first experiment, the carrier-to-noise ratio of the reflected signal decreases when the target appears, and the phase difference information changes obviously. But in the second experiment, only one of the five Beidou satellites reflected the same phase difference information. The change of carrier-to-noise ratio (CNR) information of the other four Beidou satellites is quite different from this situation. It shows that when the target appears, the intensity of the reflected signal received by the antenna is affected by the position of the target, the position of the satellite and the position of the specular point.

4. Even if the target does not appear strictly in the first Fresnel diffraction region, in a certain range, other parameters can also be used to detect the target, such as I branch component data or carrier-to-noise ratio information.

the conclusion of the second experiment is quite different from the title of the  preliminary manuscript.

These conclusions are not available in literature 25. The experimental process, data processing and conclusions of the second experiment are described in detail in the major revision.

For the above reasons, the title of the article after the major revision has changed from‘Ground target detetion utilizing the phase difference information of reflected signals from BeiDou IGSO satellite’in the preliminary manuscript  to‘Performance analysis of ground target detection utilizing Beidou satellite reflected signals’.

2. In equation (11), what’s the meaning of parameter \beta?

Response:

represents the attenuation factor of the signal. The formula is a comprehensive description of the complex changes of the reflected signal when the target exists. Whether the intensity of the reflected signal increases or decreases, it can be reflected in the phase difference information by the formula.

This part has been revised in the major revision.

3. A reasonable explanation is needed for the assumption of equation (12).

Response:

The author would like to express his gratitude to the reviewers for their opinions on this article.

The second experiment added in this paper is mainly based on the consideration of this opinion. As shown in formula 9, there are many factors affecting the intensity of the reflected signal when the target appears. But in the first experiment, because the target is fixed and positioned directly below the antenna, it is not representative.

In order to make the hypothesis of formula 12 more reasonable, in the second experiment, the experimental verification of uniform moving target was added. Through extracting the carrier-to-noise ratio (CNR) information of different satellite reflection signals, it is found that formula 12 has no errors, but only represents a special case.

The relevant information can refer to the variation of carrier-to-noise ratio (CNR) information of five different satellite reflected signals in Figure 3.

Figure 3. CNR information of five BeiDou satellite reflected signals in the second experiment: (a)PRN2 satellite; (b) PRN3 satellite; (c) PRN23 satellite; (d) PRN27 satellite; (e) PRN8 satellite.

Therefore, in the major revision, this part is supplemented and explained in detail. In this paper, the problem has been solved and added to the corresponding position of the article.

4. In equation (18), how to determine the cost factors in general? What factors

influence them?

Response:

The cost factor determines the choice of the best threshold, which is determined by the criterion of selection. In the optimal communication system, the minimum total error probability criterion, namely Bayesian criterion, is usually used, and Under this criterion, the threshold value can be expressed as:

In this paper, the problem has been solved and added to the corresponding position of the article.

5. Theoretical deduction of the performance and comparisons with other methods are required.

Response:

Thank you very much for the opinion of the reviewer. In the second experiment, the author uses I branch component data and carrier-to-noise ratio information to detect the target, and compares them with the phase difference information mentioned in the article.

If there is no reviewer's opinion, even if the second experiment is added, the author will not think of using I branch component data to detect targets, nor will he find the detection limitation of phase difference information and the relationship between the direction of target motion and I branch component data. The results can be seen in Table 1.

Table 1.  Target detection time results from the BeiDou satellite reflected signals: (A) I branch component data; (B) phase difference information; (C) carrier-to-noise ratio.

In the process of comparing phase difference information with I branch component data and carrier-to-noise ratio information, the author found new phenomena. Detailed information can refer to the answer to question 1 and the paper after major revision.

6. In Figure 7, there are eight satellites in the sky. Why just three satellites be captured and tracked?

Response:

Satellites in the sky map refer to the satellites that can be seen at the time of data acquisition. If a right-handed circularly polarized antenna is positioned vertically upward, the antenna can receive direct signals from multiple satellites. Since the polarization characteristics of the signal change after reflection, a left-handed circularly polarized (LHCP) antenna can be used to receive the reflected signal. In the first experiment, the LHCP antenna is placed nearly horizontally downward, the direction of which is west.

Figure 4. The skyplot information in the first experiment.

In the application of remote sensing technology using GNSS reflection signal, the characteristics of reflection signal are mainly forward scattering, so the reflection antenna can only receive the reflection signal of satellite located on one side of the sky direction, and the satellite reflection signal in the opposite direction can not be received by the antenna. This part can refer to the information of the star map in the first experiment, as shown in Figure 4.

7. The author concluded the probability density distribution of phase difference can

also be used as a criterion for target detection, and the presence of target can also be

judged by the CNR information of BeiDou reflected signals. However, which

observation is more appropriate?

 Response:

In the preliminary manuscript, it is proposed that the probability density function of phase difference information can be used to detect the target.

After two experiments, it is found that there is a great uncertainty in this parameter. The following is a detailed analysis of the results of two experiments.

In the first experiment, because the target is stationary, and in two fixed periods of time 10-20 and 20-30 seconds. During these two periods, the PRN9 satellite phase difference information has two obvious changes, that is, 90 degrees, and the total time length of this change is 30 seconds, accounting for about 43% of the data acquisition time (70s). By comparing the phase difference information of the Yellow ellipse position in Figure 5, we can find that the probability density distribution of the satellite phase difference information is quite different.

Figure 5. The probability density distribution of phase difference information of three satellites in the first experiment.

In the second experiment, because the target is moving at a uniform speed, it takes about 2 seconds to pass under the antenna each time. During the whole experiment, the effective time of the target passing under the antenna is 28 seconds, accounting for about 15% of the total acquisition time (180s). Compared with the Yellow position in Figure 6, only PRN8 satellite is different from other satellites, but the change is not obvious as shown in Figure 6. Other satellites can also detect targets using I branch component data, phase difference information or carrier-to-noise ratio information, but the probability density distribution can not accurately determine whether the target exists or not.

Therefore, after the experimental verification, the viewpoint of using probability density distribution as detection parameters mentioned in the preliminary draft of this paper is deleted, and the new detection parameters are proposed, which include I branch component data, phase difference information and carrier-to-noise ratio information. The three new parameters were validated by the second experimental data.

According to the above reasons, the analysis of probability density is deleted in the major revision.

Figure 6.  Probability density distribution of phase difference information of five Beidou satellites in the second experiment.

8. The theoretical analysis of the second part is unclear, for example, the formula

9-11.

Response:

The theoretical deduction of this part has two main purposes:

1. The difference between antenna receiving reflector antenna in the presence and absence of contrast target.

2. What’s the reason to reflect the change of reflected signal caused by the existence of target in phase difference information ?

The position of the target is also closely related to the intensity of the reflected signal, referring specifically to the PRN9 satellite in the first experiment and the PRN satellite in the second experiment. When the target is near the mirror point of the satellite and the azimuth angle of the satellite is close to the direction of the antenna, the carrier-to-noise ratio will decrease due to the change of several main factors in formula 9, and the reflected signal will attenuate obviously. This feature is reflected in the change of phase difference information through formula 11.

In order to illustrate the change of carrier-to-noise ratio of the reflected signal in Figure 3, the corresponding part of the article has been modified. Limited by the time relationship, the detailed theoretical model will continue to improve in the future work.

However, the theoretical analysis of this part is more complex, and can only be analyzed by several main factors in the paper.

Influencing factors can not be quantitatively analyzed at present, which will need to establish the model to fully explained.

9. The experiment is too simple. If the target is always there, what will be the results? And if the goal is big enough to cover the LHCP antenna’s footprint, what will

happen?

Response:

This opinion is divided into two parts to respond:

1. The experiment in the preliminary manuscript is really simple, and there is no new conclusion. Therefore, in this overhaul, the second experiment was added, and the fixed target in the first experiment was replaced by moving, and new conclusions were obtained. For more information, please refer to the response in Opinion 1 and the revised paper.

2. If the goal is big enough to cover the LHCP antenna’s footprint,  which needs to be discussed in two parts:

Ø If the target is in a moving state, it can also be effectively detected by the results of the second experiment in this paper, which can be seen in Table 1.

Ø If the target is stationary and can cover the footprint of the left-handed antenna, the target can not be detected effectively in this case. Because the most fundamental principle of target detection using GNSS reflection signal is that when the medium of the reflection surface is different, it can be extracted by the characteristics of the reflection signal, so that different media can be distinguished. This phenomenon is also applicable to sea ice detection, oil spill detection, forest cover and other applications. If the dielectric properties of the reflecting surface are identical, it can not be effectively judged by the characteristics of the reflecting signal.

10. There are many grammar and typo errors. For example, “detetion” in the title

“target” in formula 13, etc.

Response:

   In order to avoid similar grammatical problems of the major revision, besides careful examination by the author, the paper are also submitted to the MDPI language review body for language editing. The certificate is shown in Figure 7.

Figure 7. The certificate of English language correction by MDPI.

In a word, according to the reviewer's opinion, this paper adds a second supplementary experiment and carries out data processing. According to the new conclusion of the second experimental result, it is quite different from the conclusion in the preliminary manuscript. The new conclusions and related results are described in detail in the major revision.

Finally, it is hoped that the reviewers can make valuable opinions on the overhauled paper , which can promote the research of this technology. Thank you very much.

Author Response

Dear reviewers: 

First of all, I am very grateful to the reviewers for their valuable opinions on the preliminary manuscript. Using Beidou reflection signal to detect small targets on the ground is a new application direction. Both theoretical research and experimental verification are new explorations of this technology. Because of this, there are also many problems, which need reviewers to provide you with good views and improvements on the problems and shortcomings.

According to the reviewer's opinion, this paper has been revised a lot. In order to better answer the important opinions put forward by the reviewers and to remedy the problems in the first experiment, the author designed and completed the second improvement experiment on April 04, 2019. At the same time, the paper framework and main research contents were rewritten.

1. Title: Please, check English.

Response:

In the preliminary manuscript, there are grammatical errors in the title, which is ‘Ground target detection utilizing the phase difference information of reflected signals from BeiDou IGSO satellite’.

In order to avoid similar grammatical problems of the major revision, besides careful examination by the author, the paper are also submitted to the MDPI language review body for language editing. The certificate is shown in Figure 1.

In addition, according to the content after the major revision, the title of the paper has also been revised.

In the preliminary manuscript, we mainly discuss how to use the phase difference information of Beidou IGSO satellite for target detection, but there is no new conclusion in the paper, so the second supplementary experiment is carried out. The experiment was conducted on April 4, 2019. The experimental scenario is shown in Figure 2.

Figure 1. The certificate of English language correction by MDPI.

Figure 2. Scenario of the second experiment.

In this experiment, the movement of the target is pulled by a rope and moved back and forth in the East-West direction. The total of acquisition time is 3 minutes. In this process, the target passes through the antenna for 14 times.

According to the results of the second experiment, there are several new findings as follows:

1. When the target is moving, the satellite in different azimuth has different time to detect the target, and the detection parameters are also different, which does not depend solely on the phase difference information.

2. The changing trend of I branch component data of individual satellites is closely related to the moving direction of the target, such as the PRN2,PRN 23 and PRN27 in Figure 3.

Figure 3. I branch component of five BeiDou satellite reflected signals: (a) PRN2 satellite; (b) PRN3 satellite; (c) PRN8 satellite; (d) PRN23 satellite; (e) PRN27 satellite.

3. In the first experiment, the carrier-to-noise ratio of the reflected signal decreases when the target appears, and the phase difference information changes obviously. But in the second experiment, only one of the five Beidou satellites reflected the same phase difference information. The change of carrier-to-noise ratio (CNR) information of the other four Beidou satellites is quite different from this situation. It shows that when the target appears, the intensity of the reflected signal received by the antenna is affected by the position of the target, the position of the satellite and the position of the specular point.

4. Even if the target does not appear strictly in the first Fresnel diffraction region, in a certain range, other parameters can also be used to detect the target, such as I branch component  data or carrier-to-noise ratio information.

Because the conclusion of the second experiment is quite different from the title of the preliminary manuscript.

For the above reasons, the title of the article after the major revision has changed from‘Ground target detetion utilizing the phase difference information of reflected signals from BeiDou IGSO satellite’in the preliminary manuscript  to‘Performance analysis of ground target detection utilizing Beidou satellite reflected signals’.

2. Line 8: “phase difference information”, what are “the lags” used to calculate this difference?

Response:

The calculation of phase difference information is mainly completed in the process of acquisition and tracking of Beidou satellite reflection signal.

The phase difference information is to measure the difference of frequency and phase between the received signals and the locally generated replicas, which can be expressed as follows:

Where  is Q branch component data and is I branch component data.

3. The introduction must be improved. Some comments are not relevant for this work. On the other hand, some relevant papers about “GNSS-R+PLL+Multiconstellation+target detection” are not included in the references e.g.:

Response:

According to the reviewers' opinions and suggestion, the author focuses on the following articles and adds the relevant research results to the new paper:

1. Carreno-Luengo H, Camps A. First Dual-Band Multiconstellation GNSS-R Scatterometry Experiment Over Boreal Forests From a Stratospheric Balloon[J]. IEEE Journal of Selected Topics in Applied Earth Observations & Remote Sensing, 2016, 9(10):4743-4751.

(In this paper, the PYCARO reflectometer, which flew on-board a stratospheric balloon, has been exploited to measure the scatterers’ height fluctuations using the phase of the peak of reflected complex waveforms. Besides that, the difference of coherent-to-incoherent scattering ratio over boreal forests and lakes is also discussed. )

2. H. Carreno-Luengo, S.T. Lowe, C. Zuffada, S. Esterhuizen, and S. Ovesigharan. “GNSS-R from the SMAP and CyGNSS missions: First Assessment of Polarimetric Scatterometry and Ocean Altimetry," In Proc. IGARSS.2017.

(In this paper, the signal scattering over ocean surface from GPS satellites have been discussed . In the case of smooth surface, the signal-to-noise (SNR) ratio and phase information will follow the Fresnel coefficients. In the case of rough surface, the signal scattered similarly at vertical polarization and horizontal polarization. Based on it, the results of polarimetric ratio over the sea ice has good performance. )

3. Carreno-Luengo H, Lowe S, Zuffada C, et al. “Spaceborne GNSS-R from the SMAP mission: first assessment of polarimetric scatterometry over land and cryosphere," Remote Sens.Vol 9,no 4,2017.

(In this paper, the scattering properties over land have also been evaluated using the SNR information, the polarimetric ratio, and the width of the waveforms’ trailing and leading edges .)

4. Carreno-Luengo H , Camps A , Querol J , et al. “First Results of a GNSS-R Experiment From a Stratospheric Balloon Over Boreal Forests," IEEE Trans. Geos. Remote Sens.Vol 54,no 5,2016,pp:2652-2663.

(In this paper, a strong coherent component in the forward scattered signal have been proved from the results during the Balloon EXperiments for University Students (BEXUS) 17 stratospheric balloon. According to the multimodal behavior of received power, the coherent scattering component from the forest is analyzed. )

5. Fabra F, Cardellach E, Rius A, et al. “Phase Altimetry With Dual Polarization GNSS-R Over Sea Ice," IEEE J. 497 Sel. Top. Appl. Earth Obs. Remote Sens.Vol 50,no 6,2012,pp:2112-2121.

  (In this paper, The variability of coherent phase samples and polarimetric measurements are compared with in situ observations to make a realistic rough characterization of the ice cover , the reflected signals of which can monitor the complete process of sea ice formation and melting of Greenland during a 7-month period in 2012.)

   New references have been added to the introduction section of new paper.

4. Line 23: “Due to higher angular velocities, it can not provide stable geometric relationships for GNSS-R technique”. This point should be better justified.

Response:

According to the reviewer's opinion, the problem has been corrected. Based on the constellation structure of Beidou system, the description of this sentence is changed to:‘The BeiDou system can not only provide the signal of the MEO satellite, but also the signals of the inclined geosynchronous orbit (IGSO) and the geostationary Earth orbit (GEO) satellites, which can provide stable geometry and better coverage capability in the mid- and low-latitude regions.

5. Line 30: “The relationship between coherent tome of reflected …”. What does “coherent time” mean?

Response:

Based on the reviewers'opinions, the author reviews the following literature:

Li W, Fabra F, Yang D, et al. “Initial Results of Typhoon Wind Speed Observation Using Coastal GNSS-R of BeiDou GEO Satellite ," IEEE J. Sel. Top. Appl. Earth Obs. Remote Sens.Vol 9,2016, pp:4720-4729.

In the preliminary manuscript, the narrative of this part is not rigorous. After amendment, the description of this part is changed to:

‘The relationship between reflected waveform parameters, such as coherent time, and the ocean wind speed were analyzed.’

6. Line 38: “the concentration of sea ice is correlated with the polarization ratio…”. This result is found in the literature e.g.:

Response:

Based on the reviewers'opinions, the author reviews the following literature:

Fabra F, Cardellach E, Rius A, et al. “Phase Altimetry With Dual Polarization GNSS-R Over Sea Ice," IEEE J. 499 Sel. Top. Appl. Earth Obs. Remote Sens.Vol 50,no 6,2012,pp:2112-2121.

In the introduction part of the preliminary manuscript, the related literature refers to the technology of detecting sea ice by using the Beidou reflection signal, and also refers to the polarization ratio information mentioned in this paper. Therefore, the research completed in this paper is mainly described as follows:

‘The variability of the coherent phase samples and polarimetric measurements was compared with in situ observations to make a realistic rough characterization of the ice cover. The signals reflected  from this were used to monitor the complete process of sea ice formation and melting of Greenland during a 7 month period in 2012.’

7. Line 44: What is the difference between target and ground media?

Response:

In the paper, the target's reflective medium is metal, while the ground's reflective medium is concrete. Under the excitation of L band, the difference between the two mediums lies in the reflectivity of the signal, which is like the difference between sea water and sea ice in the sea ice detection using GNSS reflected signals.

8. Line 47: What are the “unique advantages of Beidou System in the field of remote sensing”?

Response:

The special advantage of Beidou system mentioned in the preliminary manuscript is that the Beidou GEO satellite can provide very stable geometric relationship in the field of remote sensing using GNSS reflected signals. However, this feature of GEO satellite is not used in this paper, but the performance of all Beidou satellites is discussed. Therefore, this part of the statement has been deleted in the major revision.

9. Section 2: It is recommended to add more comment relevant to GNSS-R, with focus in this particular work.

Response:

Thank you very much for your opinions. Ground target detection using Beidou reflection signal is a branch of the field of reflection signal remote sensing. In this part of the theoretical description, the introduction related to reflection should be added.

According to the opinions of the reviewers, the author has consulted in detail the literature on the application of remote sensing of reflected signals, and has a certain grasp of the theory of this part.

But the main content discussed in this paper is the application of phase difference information and carrier-to-noise ratio information in target detection, so it is necessary to introduce the theory of phase difference information. Especially in the second experiment, two new phenomena were found: 1. The change of carrier-to-noise ratio (CNR) information of different satellites when the target moves to different positions; 2. The relationship between I branch component data and the direction of target motion. All these phenomena need to be explained by a special theoretical model.

Limited by the time relationship of the major revision, the author intends to establish relevant theoretical models in the future research, and explain the phenomena in this paper from the perspective of reflected signals.

I'd like to know whether the reviewer agrees this response to the opinion.

10. Line 123: Could the authors describe these “requirements for the surrounding environment”?

 Response:

According to the reviewer's opinion, this part is explained as follows and has been added to the corresponding part of the paper:

 ‘Because this experiment is a preliminary verification of the technology, if the surrounding environment is complex, the reflected signal received by the antenna
will have interference, which will affect the analysis of the results.’

11. Line 166: “the first glistening zone …”. There is glistening zone” and “first, second etc Fresnel zones”. Check.

 Response:

Thank you very much for your opinions on this article. This description in the original article is really incorrect.

In the second experiment, even if the target is not in the first Fenier diffraction region, the antenna can still receive the reflected signal of the target. The difference is that the intensity of the reflected signal changes. For more information, see the results of the second experiment in this paper.

According to the opinions of the reviewers, this part has been revised.

12. Line 176: “the reflected signals received from these two BeiDou satellites were just reflected from the road surface, not from the target surface”. How the authors could justify that being PRN 16 and 24 out of the antenna footprint, the GNSS receiver is able to collect reflected signals (Fig. 13)? Actually, the received power increases for PRN 24, while it deceases for PRN 16. How the authors can justify this observation (Figure 13)?

Response:

The author is very grateful to the reviewers for this opinion. Based on it, in order to give a reasonable explanation for this phenomenon, this paper conducts a second verification experiment and find the new phenomenon.

In the second verification experiment, the target moves uniformly in the East and west, and passes through the antenna's position 14 times. The carrier-to-noise ratio (CNR) of five signals is different by processing the collected signals, as shown in Figure 4.

 In the second experiment, because the target is moving, it also shows that even if the target is not in the first Fresnel diffraction region, the effective information of the target can be observed from reflected signals of different satellites.

Therefore, this part has been revised in the paper after overhaul.

Figure 4. CNR information of five BeiDou satellite reflected signals in the second experiment: (a)PRN2 satellite; (b) PRN3 satellite; (c) PRN23 satellite; (d) PRN27 satellite; (e) PRN8 satellite.

13. Lines 179, 185: These points should be better explained.

Response:

Thank you very much for the opinion of the reviewers. In the preliminary manuscript, this part is written as follows:

‘When the target appears in the coverage of antenna, the phase difference information of reflected signal is kept near 90 degrees and the duration of the phenomenon depends on the existence time of the target.’

The narrative of this part is not rigorous. Especially in the second experiment, the target appeared in the area covered by the antenna. Only the phase difference information of PRN8 satellite appeared similar phenomenon (Figure 4(e)), but the reflected signals of the other four satellites did not appear.

Combined with the phase difference information of PRN9 satellite in the first experiment, it is shown that this phenomenon can only occur when the target is in a specific position with the specular point of Beidou satellite .

  In the revised article, the following corrections have been made:

 ‘If the target is located in the first Fresnel diffraction region of the satellite when the direction of LHCP antenna is in the azimuth of the satellite, the phase difference information of reflected signal is kept near 90 degrees, and the duration of the phenomenon depends on the existence time of the target. ’

14. Line 187: Why “20 degrees” is the normal level?

Response:

The conclusion is based on data statistics.

In the first experiment, PRN9 satellite is taken as an example to extract the phase difference information when the target is not put in, and the statistical analysis is carried out. The results are shown in Figure 5.

According to Figure 5, it can be found that when there is no target, the phase difference information of the reflected signal concentrates within 20 degrees, while the phase difference information greater than 20 degrees occupies a small proportion. The information of phase difference greater than 25 degrees is almost zero. The phase difference information of other satellites also has the same phenomenon, so the 20 degree range is regarded as a normal range of the phase difference information of the reflected signal when there is no target.

Figure 5.  Analysis of Phase Difference Information Distribution

of Reflected Signal without Target.

15. Line 204: “This phenomenon is caused by the influence of the signal when people place metal boxes”. This comment is not justified. Based on this hypothesis, it could be argued the same for PRN 9. As such, this work could be wrong. This issue could be the most important error in this work. The experiment should be performed again to check the results without the potential influence of people.

Response:

Thank you very much for this opinion, which is the most important reason for the second experiment.

In order to solve this problem, the author improves it on the basis of the first experiment. In the second experiment, the target was moved by two ropes to avoid the impact of the human placement of the box.

The result of PRN8 satellite is the same as that of PRN9 in the first experiment, as shown in Figure 6-7. Therefore, this phenomenon is not caused by human placement, but by the existence of the target.

Figure 6. Phase difference information of PRN8 satellite

in the second experiment

Figure 7. Carrier-to-noise ratio information of PRN8 satellite

in the second experiment

16. Line 215: The authors claim that they are performing “probability density analysis...”, but they only include the histograms. This information is redundant with previous figures. Please, improve the analysis.

Response:

In the preliminary manuscript, it is proposed that the probability density function of phase difference information can be used to detect the target.

After two experiments, it is found that there is a great uncertainty in this parameter. The following is a detailed analysis of the results of two experiments.

In the first experiment, because the target is stationary, and in two fixed periods of time 10-20 and 20-30 seconds. During these two periods, the PRN9 satellite phase difference information has two obvious changes, that is, 90 degrees, and the total time length of this change is 30 seconds, accounting for about 43% of the data acquisition time (70s). By comparing the phase difference information of the Yellow ellipse position in Figure 8, we can find that the probability density distribution of the satellite phase difference information is quite different.

Figure 8. The probability density distribution of phase difference information of three satellites in the first experiment

In the second experiment, because the target is moving at a uniform speed, it takes about 2 seconds to pass under the antenna each time. During the whole experiment, the effective time of the target passing under the antenna is 28 seconds, accounting for about 15% of the total acquisition time (180s). Compared with the Yellow position in Figure 9, only PRN8 satellite is different from other satellites, but the change is not obvious as shown in Figure 9. Other satellites can also detect targets using I branch component data, phase difference information or carrier-to-noise ratio information, but the probability density distribution can not accurately determine whether the target exists or not.

Therefore, after the experimental verification, the viewpoint of using probability density distribution as detection parameters mentioned in the preliminary draft of this paper is deleted, and the new detection parameters are proposed, which include I branch component data, phase difference information and carrier-to-noise ratio information. The three new parameters were validated by the second experimental data.

According to the above reasons, the analysis of probability density is deleted in the major revision.

Figure 9.  Probability density distribution of phase difference information of five Beidou satellites in the second experiment.

17. Please, check carefully every point before any comment on Section 5. In the present form, no conclusions can be derived from this work.

Response:

Because the first experiment is relatively simple, there is really no new conclusion to be drawn according to the results of the first experiment.

By comparing the results from the two experiments, the following conclusions can be drawn,which has been added to the major revision.

1. When the target is completely in the first Fenier diffraction zone and in the footprint of the reflector antenna, the carrier-to-noise ratio of the BeiDou satellite reflector signal decreases, and it can be detected through the I branch component data and carrier-to-noise ratio information (as shown for the PRN9 satellite in the first experiment and the PRN8 satellite in the second experiment);

2. When the target moves to different positions, the carrier-to-noise ratio of the BeiDou satellite reflection signal does not decrease due to the relative change between the positions of the target and the mirror point. When the mirror position does not appear directly below the antenna but is offset, and the carrier-to-noise ratio information of the BeiDou satellite reflection signal increases, the target can be effectively detected by using I branch component data and carrier-to-noise ratio information, but there is no obvious change in phase difference information, as shown for the results of the PRN3 satellite in the second experiment.

When the offset distance increases, the target may appear in the second or third Fenier diffraction region of the satellite, and the reflected signal from the target can still be received. When the target moves, the change in the reflected signal is more complex than the two cases mentioned above, and a decrease and increase of the reflected signal occurs. The order of the reflected signal is related to the direction of the target motion and the position of the mirror point. In this case, the effective information of the target can be seen through the I branch component data and the carrier-to-noise ratio information;

3. In the second experiment, we can see the range of target detection using the BeiDou satellite reflection signals. Taking the second experiment in this paper as an example, the effective detection range was about 1 m in the antenna position, a range which was not available in the first experiment.

4. For the PRN23, PRN27 and PRN2 satellites, the movement direction of the target can also be related to the changes in the I branch component data, which can be seen in Figure 3.

In a word, according to the reviewer's opinion, this paper adds a second supplementary experiment and carries out data processing. According to the new conclusion of the second experimental result, it is quite different from the conclusion in the preliminary manuscript. The new conclusions and related results are described in detail in the major revision.

However, there are still several problems to be solved in this paper:

1. It is unclear why the intensity of the reflected signal increases when the target appears. At present, the author has not established a new theoretical model to fully explain the occurrence of this phenomenon.

2. in the second experiment, the detection range was in the direction of the target's motion, that is, about 0.5 m before the tripod with the antenna, and the detection range at other distances was not discussed.

3. There has been no detailed theoretical explanation for the relationship between the direction of target motion and the changes in I branch component data.

4. The effective detection thresholds of different detection parameters have not been discussed in detail.

These problems will be studied in the future and new research results will be published in time.

Finally, it is hoped that the reviewers can make valuable opinions on the overhauled paper , which can promote the research of this technology. Thank you very much.

Round  2

Reviewer 1 Report

The authors have made major revisions according to the comments. I have no further comments.

Author Response

Dear  reviewer:

 Thank you very much for your valuable comments on this paper. Because of your suggestion, the quality of the paper has been greatly improved. 

Best wishes! 

Dr.Gao

Reviewer 2 Report

The authors have improved the paper and now it is ready for publication after a minor revision. This reviewer suggests to check the analysis and interpretation of the results, trying to be more synthetic when possible.

Author Response

Dear  reviewer:

Thank you very much for your valuable comments on this paper. Because of your valuable suggestions, the quality of the paper has been greatly improved.

According to your suggestion, the analysis and interpretation of the results  has been checked and adjusted. The minor revision is completed.

Best wishes!

Dr.Gao